# High-Intensity Focused Ultrasound in Dermatology: A Review with Emphasis on Skin Cancer Management and Prevention

**DOI:** 10.3390/cancers17213518

**Published:** 2025-10-31

**Authors:** Bartosz Woźniak, Piotr Sobolewski, Natalia Sauer, Mateusz Koper, Jacek Calik

**Affiliations:** 1Old Town Clinic, 50-136 Wroclaw, Poland; 2Dermatology Clinic, National Institute of Medicine of the Ministry of Interior and Administration, 02-507 Warsaw, Poland; 3Department of Clinical Pharmacology, Faculty of Pharmacy, Wroclaw Medical University, 50-556 Wroclaw, Poland; natalia.sauer@umw.edu.pl

**Keywords:** high-intensity focused ultrasound, skin cancer, basal cell carcinoma, actinic keratosis, non-invasive therapy, field cancerization, dermatologic oncology, 20 MHz HIFU, skin surgery alternative

## Abstract

**Simple Summary:**

Skin cancers are becoming increasingly common, and many patients need treatments that are both effective and minimally invasive. High-intensity focused ultrasound (HIFU) is a new technique that uses concentrated sound waves to precisely destroy abnormal skin cells without cutting the skin. This review explains how HIFU works and summarizes the current evidence for its use in skin conditions such as actinic keratosis and basal cell carcinoma. Early studies suggest that HIFU can remove precancerous and cancerous lesions with acceptable safety and promising clinical results. Because it can target only the affected area while leaving healthy skin intact, HIFU may become an important alternative or addition to surgery and other established methods. The findings presented here may help guide future research and clinical practice in non-invasive skin cancer management.

**Abstract:**

High-intensity focused ultrasound (HIFU) has recently emerged as a novel non-invasive treatment modality in dermatology, offering precise ablation of cutaneous lesions with minimal damage to surrounding tissue. Originally developed for deep-seated tumors, dermatological HIFU platforms operating at ~20 MHz enable submillimeter-scale treatment of thermal or mechanical injuries localized to the epidermis and superficial dermis, making them suitable for managing benign, premalignant, and malignant skin conditions. This review outlines the mechanistic basis of HIFU—including thermal coagulation, acoustic cavitation, and immunomodulatory effects—and presents the current evidence for its efficacy in treating actinic keratoses and basal cell carcinomas (BCCs), where early studies report clearance rates of 70–97% and excellent cosmetic outcomes. Compared to conventional therapies such as surgery, photodynamic therapy, or cryotherapy, HIFU offers reduced procedural pain, faster healing, and the ability to treat multiple lesions in a single session. Its role in field cancerization and potential utility in prophylaxis for high-risk skin areas are also explored. While promising, long-term oncologic outcomes and standardized treatment protocols remain under investigation. HIFU represents a significant advancement in non-invasive skin cancer management, aligning oncologic efficacy with patient-centered care.

## 1. Introduction

Non-melanoma skin cancers (NMSCs), particularly basal cell carcinoma (BCC) and squamous cell carcinoma (SCC), are the most prevalent malignancies worldwide. Their incidence continues to rise globally due to factors such as population aging, increased ultraviolet (UV) radiation exposure, and changing environmental and behavioral patterns [1]. BCC accounts for approximately 70–80% of all NMSCs and is generally characterized by slow growth and local invasiveness, rarely metastasizing. In contrast, SCC comprises 20–25% of NMSCs but has a higher potential for local invasion and metastasis, making it responsible for the majority of NMSC-related deaths [2,3].

BCC and SCC share UV exposure as a common etiological factor, but their molecular pathogenesis diverges significantly. BCC is primarily driven by constitutive activation of the Hedgehog signaling pathway. Mutations in PTCH1, SMO, and SUFU are observed in up to 85% of BCC cases, leading to downstream GLI transcription factor activation and unregulated keratinocyte proliferation [4]. Additional mutations in TP53 and other tumor suppressors may contribute to disease progression. In contrast, SCC arises through cumulative UV-induced DNA damage, with a high mutational burden and characteristic UV-signature mutations. TP53 mutations are nearly ubiquitous in SCC and are often among the earliest events [5]. Moreover, loss-of-function mutations in NOTCH1 and NOTCH2, which regulate epidermal differentiation, are seen in over 70% of SCCs, underscoring their critical tumor suppressor roles [6].

While surgical excision remains the gold standard for treating localized NMSCs, it is not without limitations. Repeated surgeries in cosmetically or functionally sensitive areas such as the eyelids, ears, or nose can lead to scarring, disfigurement, or impaired function. Furthermore, elderly patients or those burdened with multiple comorbidities are frequently deemed unsuitable for surgical intervention. In this context, there is an increasing emphasis on treatment modalities that not only ensure oncological efficacy but also align with the principles of patient-centered care, prioritizing functional preservation and aesthetic integrity [7]. While current non-invasive approaches such as photodynamic therapy or topical immunomodulators provide viable options for select cases, their efficacy is often limited by factors such as lesion thickness, patient adherence, or procedural discomfort.

One such promising modality is high-intensity focused ultrasound (HIFU) [8]. HIFU delivers focused acoustic energy to induce controlled thermal coagulative necrosis within the target tissue, without damaging the overlying skin. Recent studies using 20 MHz HIFU for basal cell carcinoma (BCC) have demonstrated high efficacy, excellent cosmetic outcomes, and minimal patient discomfort. Treated lesions typically exhibit immediate whitening and edema, indicative of effective ablation, followed by rapid healing with minimal scarring. In a prospective cohort, no tumor recurrences were observed at six months follow-up, suggesting a durable and long-lasting therapeutic response. HIFU offers several additional advantages that distinguish it from conventional treatment modalities. The procedure is entirely non-invasive and does not require the use of local anesthesia or intralesional injections, which significantly enhances patient comfort during the intervention. Moreover, the integration of real-time dermoscopic monitoring enables precise visualization of tissue response, allowing clinicians to adjust treatment parameters dynamically and ensure complete lesion coverage. The post-treatment adverse effects are generally mild and transient, limited to short-lived erythema, edema, or crust formation, while complications such as pigmentation disorders or scarring are rare and of negligible clinical significance. A key strength of HIFU lies in its ability to tailor energy delivery to the specific depth of the lesion, as determined by pre-procedural ultrasonographic assessment, thereby maximizing treatment precision and minimizing the risk of collateral tissue damage. This precision makes HIFU particularly suitable for lesions located in cosmetically sensitive areas, such as the face, where preserving the integrity of surrounding healthy skin is crucial. Patients treated with HIFU report high levels of satisfaction, appreciating both the minimally invasive nature of the procedure and the superior aesthetic outcomes compared to conventional surgical excision or cryotherapy. The combination of efficacy, safety, precision, and patient-centered cosmetic benefits underscores HIFU’s potential as a valuable alternative in the therapeutic arsenal for BCC management.

In parallel, advances in radiation-based modalities have broadened the non-invasive treatment arsenal. Conventional external beam radiotherapy (EBRT) has long been used for NMSC management, especially in inoperable patients or those with high-risk features. However, one of the most innovative developments is the integration of three-dimensional (3D) printing with high dose rate (HDR) brachytherapy. Brachytherapy involves placing a radioactive source close to or within the tumor, delivering localized radiation with minimal exposure to adjacent healthy tissue [9].

The precision of brachytherapy has been significantly enhanced by 3D-printing technology. Patient-specific applicators can now be custom-designed based on CT or surface imaging to conform perfectly to irregular anatomical contours. This customization minimizes air gaps between the applicator and skin, improves dose homogeneity, and allows accurate repositioning during multiple treatment sessions [10]. In a recent study, 3D-printed applicators achieved submillimeter placement accuracy and air gaps under 0.5 mm, resulting in reliable dose delivery and consistent tumor coverage [9]. Clinical outcomes in patients with inoperable facial BCC and SCC showed complete remission in all cases following a 51 Gy HDR protocol using 3D-printed applicators, with excellent cosmetic results and no significant toxicity [11].

Moreover, dosimetric analyses comparing commercial treatment planning systems and independent software have confirmed the precision and reproducibility of 3D-guided HDR brachytherapy. Parameters such as D90, V100, and conformity indices demonstrated strong correlation, validating the accuracy of dose delivery and ensuring quality assurance in clinical settings [12]. This technology not only enhances treatment efficacy but also reduces operator dependency and improves patient comfort.

Complementing these modalities, topical therapies such as imiquimod 5% cream offer non-invasive options for select superficial and nodular periocular BCCs. Imiquimod activates the toll-like receptor 7 (TLR7), triggering a local immune response and tumor regression. A recent systematic review reported an 82% clearance rate and excellent cosmetic outcomes, especially in patients where surgery was contraindicated [13].

In summary, NMSCs remain a significant public health challenge due to their rising incidence, recurrent nature, and the limitations of surgical management. A deeper understanding of their molecular underpinnings has paved the way for novel, mechanism-based interventions. Among emerging therapies, HIFU stands out for its non-invasiveness, precision, and excellent aesthetic outcomes, particularly in BCC treatment. Simultaneously, the integration of 3D printing into HDR brachytherapy marks a paradigm shift in radiation-based skin cancer treatment, offering individualized, reproducible, and anatomically adaptive approaches. Together with topical immunomodulators, these advances point toward a future where NMSCs can be effectively managed through tailored, patient-centered, and minimally invasive strategies.

To guide the present review, we performed a structured PubMed search using the terms “high-intensity focused ultrasound” AND “skin” OR “dermatology” OR “actinic keratosis” OR “basal cell carcinoma” (last accessed: 30 July 2025). However, the number of available original studies and retrospective clinical cohorts—particularly those using dermatologic HIFU platforms at ~20 MHz—remains limited. Given the insufficient quantity and heterogeneity of high-quality clinical data, the application of a systematic review methodology was not feasible. We therefore adopted a narrative review format, allowing for integrative analysis of mechanistic insights, clinical outcomes, and comparative positioning within the broader therapeutic landscape.

The specific aim of this review is to summarize the mechanistic principles, current dermatological applications, and clinical evidence of HIFU, with particular emphasis on its emerging role in the management and prevention of skin cancers.

## 2. Mechanism of HIFU in Dermatology

HIFU operating at 0.5–3 MHz has been established as an effective, non-invasive modality for ablating deep-seated malignancies in major organs, treating bone metastases, and managing various intracranial disorders. Real-time magnetic resonance or ultrasound guidance enables precise placement of the acoustic focus several centimeters beneath the body surface, where energy deposition raises tissue temperatures to about 65 °C—high enough to induce coagulative necrosis in the target while preserving overlying and adjacent structures [14,15,16,17,18,19,20].

Preclinical work has shown that when high-frequency HIFU transducers are used with short focal lengths, they can repeatedly confine heat to near-surface targets within a medium [21]. Across the therapeutic window studied, the biological response varies linearly with the applied acoustic energy, permitting the fine-grained dose control required in clinical practice [22].

These properties render high-frequency HIFU particularly pertinent to clinical and cosmetic dermatology. Because the focal volume produced by an HIFU transducer decreases as the operating frequency increases, elevating the frequency enhances spatial precision. For cutaneous applications, the acoustic focus must therefore be accurately located within the epidermis, dermis, or subcutis, depending on the intended intervention. Preclinical evidence indicates that a center frequency of about 20 MHz is necessary to achieve this level of target specificity in skin [21].

Most commercially marketed body-contouring and facial rejuvenation devices—designed to act on comparatively large tissue volumes located in the subcutaneous layer or deeper—operate at ultrasound frequencies of roughly 4–10 MHz [23,24]. This frequency range does not provide the spatial precision required for highly localized treatments confined to the epidermal or dermal layers [25,26].

HIFU at 20 MHz mediates tissue destruction through two principal pathways—thermally induced coagulative necrosis and mechanically driven cavitation—while also potentially initiating an immunological response, each of which will be detailed in the subsequent sections. Figure 1 provides a concise schematic summary of these mechanisms in dermatological HIFU, illustrating the focused acoustic energy conversion into heat and cavitation, and highlighting potential secondary immune-mediated effects.

Thermal mechanisms of HIFU involve the delivery of convergent acoustic energy, which elevates the temperature at the focal spot to above 60 °C within milliseconds, resulting in instantaneous coagulative necrosis of the targeted tissue compartment [27]. Because the ultrasonic beam is sharply focused, the thermal gradient across the lesion margin is exceedingly steep—often spanning only a few cell layers—and thereby safeguarding adjacent structures and offering a decisive advantage in cosmetically sensitive cutaneous sites. Sub-ablative exposures (≈42–55 °C) can instead produce controlled hyperthermia, transiently increasing plasma—membrane permeability and activating heat shock protein-mediated stress pathways that may potentiate adjuvant therapies [27,28].

Mechanical mechanisms of HIFU involve the delivery of pulsed acoustic energy, which induces rapid pressure oscillations that nucleate and implode microbubbles (acoustic cavitation), generating intense shear forces and microjets that mechanically fractionate tissue in a process known as histotripsy [27,28]. Ancillary phenomena, including radiation force and micro-streaming, further disrupt cellular and extracellular architectures. Cavitation-driven sonoporation can transiently open nanometer-scale pores in the lipid bilayer, a feature exploited experimentally to enhance intratumoral delivery of chemotherapeutics and nanoparticles, although the present review concentrates on HIFU’s ablative applications

Immunomodulatory consequences of HIFU extend beyond direct cytotoxicity, as HIFU ablation can function as an in situ vaccine by releasing tumor-associated antigens and danger-associated molecular patterns (DAMPs). Preclinical models consistently demonstrate post-HIFU up-regulation of pro-inflammatory cytokines (e.g., IFN-γ, TNF-α) and recruitment of tumor-specific cytotoxic T lymphocytes, culminating in systemic anti-tumor immunity [27]. This dual capacity—to eradicate the primary lesion while simultaneously priming host immunity—holds particular promise for skin cancer management, and will be revisited in subsequent sections addressing combinatorial regimens and prophylactic perspectives.

Collectively, the evidence indicates that HIFU exerts its cytotoxic action through a triad of complementary processes: thermal coagulation, cavitation-driven mechanical fractionation, and secondary immunomodulation. For clinical dermatologic applications, device parameters diverge from conventional deep-tissue systems: transducers operating at 20 MHz create focal zones only a few millimeters below the skin surface, enabling selective ablation of lesions confined to the epidermis or superficial dermis while sparing underlying structures.

Precise optimization of device configuration and treatment planning is fundamental to effectively translate the mechanistic advantages of HIFU into consistent and reproducible clinical outcomes. Among the various technical variables, certain parameters exert a particularly critical influence on treatment efficacy.

One of the most salient factors is the interplay between ultrasound frequency and focal depth. Elevating the center frequency to the range of 18–22 MHz markedly reduces the acoustic wavelength, thereby confining the lateral focal diameter to approximately 0.15–0.25 mm and limiting the focal penetration depth to 1–2 mm. These dimensions closely correspond to the anatomical layers of the epidermis and papillary dermis, which are commonly involved in lesions such as actinic keratosis (AK) and superficial basal cell carcinoma (BCC) [29]. This spatial precision enables selective targeting of superficial neoplastic lesions while sparing underlying healthy tissue.

Figure 2 illustrates the inverse relationship between ultrasound frequency and focal depth, delineating the operational ranges of representative dermatologic HIFU devices. Understanding and leveraging this frequency–depth dependency is imperative for tailoring treatment protocols to the specific morphological characteristics of cutaneous lesions, thus ensuring optimal ablation precision and therapeutic outcomes.

The architecture of the acoustic pulse is another pivotal determinant in modulating the biological effects of HIFU. Continuous-wave emission facilitates maximal thermal energy deposition within the focal volume, thereby enhancing the efficacy of coagulative necrosis. In contrast, delivering microsecond to millisecond bursts at pulse repetition frequencies (PRFs) ranging from 1 to 10 Hz preferentially promotes cavitation-mediated tissue disruption, a phenomenon known as histotripsy.

Contemporary dermatologic HIFU platforms are engineered to offer real-time adjustability of pulse duration, PRF, and pulse energy, typically within the range of 0.5 to 1.5 Joules per shot for indications such as actinic keratosis (AK) and basal cell carcinoma (BCC). This level of control allows clinicians to fine-tune the balance between thermal ablation and mechanical fractionation, thereby customizing the injury profile according to lesion characteristics and therapeutic objectives. Such dynamic modulation of pulse architecture is essential for optimizing treatment precision, efficacy, and safety across diverse dermatologic applications.

Imaging and dosimetry play a critical role in ensuring the precision and safety of HIFU treatments. Pre-procedural delineation of lesion morphology is achieved using high-frequency ultrasound (20–48 MHz), which provides accurate measurements of lesion thickness and identifies critical anatomical structures in proximity to the treatment zone. This facilitates the selection of appropriate focal depths and energy parameters tailored to the individual lesion’s characteristics. Furthermore, adjunctive imaging modalities such as optical coherence tomography (OCT) and high-resolution dermoscopy enable serial monitoring of post-treatment tissue dynamics, including crust formation, desquamation, and the progression of re-epithelialization. These real-time observations support adaptive treatment strategies, allowing clinicians to refine dosimetric parameters and adjust therapeutic protocols during follow-up sessions [30]. Such iterative dose optimization is fundamental for achieving complete lesion clearance while minimizing collateral tissue injury and enhancing cosmetic outcomes.

These engineering refinements transform the general HIFU paradigm into a dermatology-tailored platform that couples submillimeter precision with real-time imaging and a favorable safety profile; these attributes are critical for managing thin cutaneous malignancies in cosmetically sensitive regions. The mechanistic framework detailed here provides the biological rationale for the clinical outcomes reviewed in the subsequent sections.

## 3. Current Dermatological Uses of HIFU (Cosmetic and Benign Applications)

Over the past fifteen years HIFU has migrated from deep-seated oncologic therapy to the dermatology clinic, where micro- and high-frequency platforms now provide incision-free treatments for skin rejuvenation, scars, and a broad spectrum of benign skin lesions. Its uptake has been driven by a favorable safety record, reproducible histological precision, and the ability to visualize or pre-select the target depth before energy delivery.

### 3.1. Aesthetic Skin Tightening and Rejuvenation

Three complementary systematic reviews now delineate the clinical performance and technical boundaries of micro-/high-intensity focused ultrasound (MFU/HIFU) for nonsurgical facial tightening. Contini et al. screened 693 records and analyzed 16 prospective studies (all-female cohorts, ≥3-month follow-up), documenting objective lifting of the brow (0.47–1.7 mm) and submental region (26–45 mm^2^ reduction) together with a pooled investigator GAIS improvement value of 92% at day 90 that persisted for one year; patient-reported GAIS values were lower (42% at day 90, 53% at day 360), and adverse effects were limited to transient erythema/edema or infrequent dysesthesia/contusions (≈2%) [31]. Extending these findings, the meta-analysis by Ling and Zhao aggregated 477 participants from 13 trials and yielded pooled GAIS responder rates of 0.77 (95% CI 0.58–0.96) and 0.69 (0.51–0.87) at 90 and 180 days, respectively; parallel patient satisfaction proportions were 0.78 and 0.71, while mean pain remained mild (3.1/10), underscoring reproducible efficacy with acceptable tolerability [32]. Crucially, Dicker et al. compared the three dominant commercial platforms—Ultherapy^®^ (Merz Aesthetics, Raleigh, NC, USA), Ultraformer III^®^ (Ugintech, Taichung City, Taiwan), and Liftera™ (Espoo, Finland)—and revealed an order-of-magnitude variability in acoustic frequency (2–10 MHz), focal depth (1.5–13 mm), per-line energy (0.1–3 J), and thermal coagulation point density; noting that only Ultherapy^®^ offers real-time imaging, the authors contend that such hardware heterogeneity likely drives the statistical heterogeneity observed across the outcome studies and advocate for device-specific reporting in future trials [33]. Collectively, these reviews support MFU/HIFU as an evidence-based option for treating mild to moderate facial laxity, delivering measurable lifting and high investigator-graded improvement with low morbidity, yet they also highlight the need for standardized dosimetry, longer-term follow-up, and inclusion of broader patient demographics before definitive treatment algorithms can be formulated.

### 3.2. Benign Cutaneous Lesions

HIFU delivers millimeter-scale, thermomechanical “pixels” of energy that can be positioned anywhere between the epidermis and mid-dermis with submillimeter precision. Thus, 20 MHz dermatologic platforms exploit this selectivity to ablate or modulate discrete foci while sparing surrounding tissue, offering an anesthesia-sparing, scar-sparing alternative to surgery, lasers, and cryodestruction for a growing list of benign skin disorders. Clinical evidence—ranging from case reports, single-center prospective series to multicenter trials—now supports its use across fibrotic, keratotic, adnexal, vascular, and inflammatory lesions, as outlined below.

#### 3.2.1. Hypertrophic Scars and Keloids

An early Bulgarian cohort (*n* = 20) treated with 4–7 MHz HIFU achieved reproducible flattening, reduced pruritus and dermal echogenicity normalization after 1–4 sessions, with no recurrences at ≤12-month follow-up [4]. Temperature mapping confirmed focal peaks of 56–65 °C limited to the high-density scar core, converting rigid collagen into a gelatinous matrix that was remodeled under post-procedural massage [34]. These data position HIFU as an adjunct or alternative to intralesional corticosteroid or laser therapy, particularly for chest wall and earlobe keloids where surgical recurrence is high. Further research is needed.

#### 3.2.2. Seborrheic Keratosis (SK)

Seborrheic keratoses are the most common benign epithelial tumors of adult skin, usually presenting as numerous lesions on the trunk, face, and extremities [5]. In an open-label pilot study of 54 lesions in 11 adults (Fitzpatrick I–II), a single pass of 20 MHz high-frequency HIFU delivered with a 0.8 mm focal depth handpiece (0.99–1.2 J, 150 ms per dose) achieved a 96% overall response rate, including complete clinical clearance in 69% of lesions [6]. Mean intra-procedural pain was mild (2–4/10 on a visual analog scale) and no serious adverse events were recorded [35].

While these early data point to high-frequency HIFU as a well-tolerated and potentially effective office-based treatment for SK, the evidence is limited to a single, small series in fair-skinned patients. Larger, controlled studies—ideally comparing HIFU with established modalities such as cryosurgery or CO_2_ laser ablation and incorporating a broader range of skin phototypes—are needed before definitive clinical recommendations can be formulated.

#### 3.2.3. Sebaceous Hyperplasia (SH)

Sebaceous hyperplasia—the benign enlargement of sebaceous glands, most commonly on the face of middle-aged and older adults—poses primarily cosmetic concerns but can mimic basal cell carcinoma, prompting intervention [36]. Conventional options (electrodesiccation, CO_2_/Er:YAG lasers, cryotherapy, isotretinoin) often trade efficacy for pain, post-inflammatory pigment change or recurrence, and therefore warrant evaluation of less invasive modalities [36].

A prospective, open-label study treated 33 facial SH lesions in six adults (five women, one man; ages 39–60 y) with a 20 MHz HIFU system [37]. A 0.8 mm focal depth probe delivered contiguous 1 J/150 ms doses at 1 mm spacing under real-time dermoscopic guidance. At the three-month assessment, 87.9% of lesions showed complete clinical clearance and the remaining 12.1% were partially reduced; no lesion was unchanged or progressive. Patients described the procedure as virtually painless, and adverse effects were limited to transient telangiectasia, mild erythema, subtle scarring, or focal hyperpigmentation at the treatment site; no serious events occurred [37].

These encouraging results are tempered by the study’s small, uncontrolled design, as well as its short follow-up and absence of skin phototype diversity. Randomized comparisons with established modalities (e.g., CO_2_ laser, electrosurgery, cryotherapy) and longer surveillance across a broader range of phototypes are needed before high-frequency HIFU can be formally recommended for routine management of sebaceous hyperplasia.

#### 3.2.4. Cutaneous Neurofibroma

Cutaneous neurofibromas are benign Schwann cell tumors that develop in up to 95% of adults with neurofibromatosis type 1 (NF1), a disorder affecting roughly 2.5 million people worldwide [38]. The hundreds of soft papules and nodules that may accumulate on visible sites produce pain, pruritus, psychosocial distress, and postsurgical scarring, yet standard management is still limited to serial excisions or ablative lasers; such options are impractical for high lesion burdens and frequently result in dyspigmentation or keloids [38]. All of these factors underscore the need for less invasive approaches.

A two-center, prospective, open-label study treated 147 cutaneous neurofibromas in 20 NF1 patients with a 20 MHz high-frequency HIFU device [39]. A 2.3 mm focal depth probe delivered 0.7 J/250 ms doses at 1–2 mm spacing under real-time dermoscopic guidance. Nine months after a single session, 48.9% of tumors showed complete or near-complete flattening on investigator assessment, and median ultrasound thickness fell by 0.53 mm; median intra-procedural pain was 3.5/10 and no serious adverse events were observed, with transient erythema and focal dyspigmentation being the most frequent effects [39]. Complementing these data, a single-patient case report documented four complete and two partial responses among seven treated lesions at 12 months when a 2.3 mm handpiece was employed, with only brief post-treatment hyperpigmentation [40].

#### 3.2.5. Benign Vascular Tumors (Cherry Angioma, Congenital Hemangioma)

Cherry angiomas (also called senile angiomas or Campbell de Morgan spots) and congenital hemangiomas are among the most prevalent cutaneous vascular tumors encountered in daily dermatological practice [41,42]. Although biologically benign, their visibility on exposed sites often precipitates cosmetic concern, while larger lesions may ulcerate, bleed, or become tender. Traditional management—electro- or hyfrecation, cryotherapy, pulsed-dye and KTP lasers, or surgical shave excisions—carries a non-trivial risk of scarring, dyspigmentation, or recurrence [43,44]. These limitations have spurred interest in non-invasive, image-guided modalities such as HIFU.

Calik et al. prospectively treated a single congenital hemangioma on the cheek (three sessions) and seven cherry angiomas on the thigh (single session) with a 20 MHz device delivering 1.1–1.2 J/150–250 ms at 0.8–1.8 mm depth [45]. At 15 weeks, the facial hemangioma displayed near-normal skin color and texture. Dermoscopy revealed a pronounced reduction in the capillary network and round junctions and an almost fully reconstituted dermal architecture; residual peripheral capillaries persisted but were virtually indistinguishable from adjacent skin. At the 12-month assessment, complete clearance with excellent cosmetic outcome persisted in five of the seven treated angiomas, whereas the remaining two showed subtle improvement. No true recurrence was documented after 1 year of follow-up [45].

The encouraging outcomes of these initial case reports warrant systematic validation. Future research should include head-to-head randomized trials comparing HIFU with established modalities such as KTP laser therapy, pulsed-dye laser, and electrocoagulation, as well as studies exploring HIFU as an adjuvant option when conventional laser treatment has plateaued and no further improvement can be achieved. Larger prospective cohorts are also required to refine dosimetry algorithms, establish long-term recurrence rates, and evaluate cost-effectiveness. Integration of high-resolution ultrasound with machine learning-supported targeting may broaden indications to mixed and arteriovenous malformations.

#### 3.2.6. Vulvar Lichen Sclerosus

Vulvar lichen sclerosus is characterized histologically by basal layer apoptosis, homogenization of the dermis, and micro-angiopathy [46]. Chronic pruritus and burning dominate the clinical picture and about 5% of women progress to squamous cell carcinoma [47]. Commercial gynecological transducers (e.g., SEAPOSTAR^®^ MODEL—CZF HIFU, Chongqing Haifu Medical Technology Co., Ltd., Chongqing, China) operate at 8–12 MHz, deliver 3.0–4.7 W acoustic power, and scan linearly at a speed of 3–5 mm/s [48]. HIFU deposits acoustic energy of 4–6 mm under the skin [48]. Controlled thermal coagulation denatures pathological collagen, stimulates neo-angiogenesis and re-epithelialization, and may down-regulate inflammatory cytokines, thereby addressing both symptoms and premalignant architecture. Objective imaging with active dynamic thermography and hyperspectral analysis confirms immediate tissue-level effects and predicts the clinical outcome [48]. Another retrospective study showed that shallower foci (2.5 mm) reduce energy requirements and lower 12-month recurrence without sacrificing efficacy [49]. Many prospective cohorts prove that overall pooled cure and reduced subjective symptoms rates average 90% at 6–12 months, with recurrence typically manifesting as localized flare amenable to re-treatment [50,51,52]. Owing to its favorable safety profile, HIFU may be incorporated into routine clinical practice as a credible alternative to the current gold standard therapy of topical clobetasol and is suitable for use in pediatric patients [53,54,55]. Robust, head-to-head randomized trials directly comparing high-intensity focused ultrasound with high-potency topical corticosteroids, calcineurin inhibitors such as pimecrolimus, fractional ablative laser platforms, and photodynamic therapy remain essential to delineate its precise position within the therapeutic algorithm for vulvar lichen sclerosus [56,57,58,59].

#### 3.2.7. Granuloma Annulare and Other Inflammatory Dermatoses

Granuloma annulare (GA) is a granulomatous disorder marked by dermal palisading histiocytes surrounding degenerated collagen [60]. Conventional modalities (topical steroids, phototherapy, cryosurgery) are frequently slow, result in scarring, or are inefficient [60]. In a 2024 prospective study of a patient with disseminated GA, 22 lesions were treated with 20 MHz dermatologic HIFU (0.3–0.6 J per exposure, 0.8–1.3 mm focal depth, 1 mm pitch) and 10 contralateral lesions received liquid nitrogen cryotherapy for intra-individual comparison [61]. All HIFU-treated plaques achieved complete or near-complete resolution within 3 months, with negligible dyspigmentation, whereas cryotherapy produced larger necrotic volumes and post-inflammatory hypopigmentation [61]. The authors postulated a “hyperthermic immunological activation” rather than pure ablative destruction, highlighting HIFU’s potential to normalize aberrant macrophage-mediated inflammation in GA [61].

Morbihan disease, a refractory variant of rosacea presenting with solid facial edema, was recently addressed in a single-case trial using macro-focused HIFU (MF-HIFU, 4 MHz, 3 mm focal depth, and intensity of 3.76 W) [62]. Three sessions produced sustained reductions in erythema, edema, and pain over 6 months without scarring [62]. Although limited to one subject, these results imply that sub-ablative thermal doses can enhance lymphatic drainage and vascular tone, complementing anti-angiogenic strategies in rosacea [62].

Although the current evidence is restricted to isolated case reports involving single patients, the favorable clinical responses and mechanistic insights obtained thus far provide a rationale—and indeed an imperative—for well-designed, prospective, randomized-controlled trials to validate the therapeutic role of HIFU across a broader spectrum of inflammatory dermatoses.

#### 3.2.8. Viral Warts

High-frequency (20 MHz) dermatologic HIFU has recently been explored as a tissue-sparing alternative for recalcitrant cutaneous warts [63]. In the first human observation, two common warts on the hand were treated with eight to ten 150 ms pulses of 1.2 J each, delivered at a nominal focal depth of 1.3 mm under real-time dermoscopic guidance [63]. The sites crusted within 48 h, re-epithelialized completely by week 3 and remained free of recurrence or scarring at the three-month follow-up [63]. The ability to confine temperatures of ≈65 °C to a submillimeter focal volume while sparing adjacent epidermis supports a single-visit strategy where keratolytics, cryotherapy, or laser ablation have failed.

A subsequent case report extended the concept to condylomata acuminate [64]. A 25-year-old man with widespread genital lesions received 5% imiquimod cream for seven weeks, achieving marked clearance of the warts. The residual lesions were then ablated in a single session with 20 MHz dermatologic HIFU, resulting in complete dermoscopic resolution by day 118 and no functional or cosmetic sequelae [64]. Crucially, the ultrasound energy produced no thermal plume or aerosol, eliminating the occupational exposure to airborne HPV particles that accompanies CO_2_ or Nd:YAG laser vaporization [64].

Nevertheless, the evidence base remains anecdotal. Controlled, dose-finding trials with clinical endpoints are required to confirm cure rates versus cryo- or laser therapy and quantify pain and anesthesia requirements at different anatomical sites. In the authors’ unpublished clinical experience, dermatologic HIFU at 20 MHz showed limited efficacy against plantar verrucae. The basis for this apparent site-specific resistance remains unclear and merits systematic investigation. Such studies will determine whether dermatologic HIFU can be incorporated into future guidelines for viral wart management.

#### 3.2.9. Healing Process After High-Intensity Focused Ultrasound Treatment of Benign Skin Lesions

A prospective series of 233 benign lesions (164 seborrheic keratoses, 37 capillary hemangiomas, 23 sebaceous hyperplasias, 9 sebaceous naevi) treated with 20 MHz HIFU revealed a stereotyped but lesion-specific repair sequence [65]. Immediately after sonication, all lesions developed focal whitening followed by thin scab formation within 48 h. Dermoscopic monitoring showed a transition to a pink reparative zone with fine arborizing vessels after 1–2 weeks. Healing times differed by histotype: flat seborrheic keratoses re-epithelialized in ≈4 weeks, whereas verrucous variants required ≈8 weeks; sebaceous hyperplasia healed in ≈3 weeks, often leaving a shallow depression; superficial vascular lesions resolved in 3–4 weeks, while deeper hemangiomas matured over 6–8 weeks with concentric fibrotic bands; and sebaceous naevi showed the slowest course, attaining complete architectural normalization at 12 weeks. No keloids, delayed ulcerations, or pigmentary scarring were observed. These data provide realistic counseling timelines, inform follow-up scheduling, and may serve as reference endpoints for future dose-finding or comparative trials [65]. Because the available evidence is restricted to a few body sites, future prospective studies should stratify healing kinetics by anatomical location and optimize parameters—focal depth, pulse width, and spot spacing—according to lesion type and anatomical site.

## 4. HIFU for Premalignant Lesions (Actinic Keratosis and Field Cancerization)

Actinic keratoses (AKs) represent early intra-epidermal keratinocyte atypia and signal a “field cancerization” of chronically photodamaged skin in which subclinical clones carry the same UV-induced driver mutations that lead to invasive cutaneous squamous cell carcinoma (SCC) [66]. The risk of progression of AK to invasive SCC is estimated to be between 0.025% and 20% for each individual lesion [67]. Although several authors maintain that individual actinic keratoses (AKs) cannot be reliably identified as lesions destined to progress into invasive squamous cell carcinoma (SCC) due to viewing them chiefly as field cancerization markers of global SCC risk [68,69], Quaedvlieg et al. described a constellation of features predictive of malignant transformation [70]. They coined the mnemonic *IDRBEU*: I, inflammation or induration; D, diameter > 1 cm; R, rapid enlargement; B, bleeding; E, marked erythema; and U, ulceration [70]. The same study also listed minor criteria (pain, palpability, pronounced hyperkeratosis, pruritus, and pigmentation) that may further raise clinical suspicion [70].

Given the dual perspective that AKs function both as markers of field cancerization and, in select cases, as site-specific precursors of invasive squamous cell carcinoma, it is essential to refine therapeutic strategies that target both the chronically sun-damaged field and the individual high-risk lesions. Field-directed modalities include topical chemotherapeutics—5-fluorouracil, imiquimod, tirbanibulin, and diclofenac—as well as daylight or conventional photodynamic therapy, fractional ablative laser resurfacing, medium-depth chemical peels, and systemic chemoprevention with nicotinamide [71]. In parallel, lesion-directed options such as liquid nitrogen cryosurgery, curettage with electrodesiccation, shave or excisional surgery, targeted photodynamic therapy, and focused ablative laser ablation remain indispensable for eradicating clinically suspicious or IDRBEU-positive plaques [71]. A comprehensive management plan that integrates both arms not only reduces overall tumor burden across the cancerization field but also addresses lesions with the greatest likelihood of malignant progression [71]. It should be remembered, however, that each of the above modalities carries its own advantages and drawbacks, and some are limited by prolonged post-treatment erythema, procedure-related pain (particularly with PDT), challenges with patient adherence, and the requisite experience and technical skill of the operator. Moreover, emerging randomized evidence shows that the subsequent risk of progression to invasive cutaneous squamous cell carcinoma can differ between treatment modalities themselves [72].

The CE-marked System ONE-M (TOOsonix A/S, Hørsholm, Denmark) exemplifies current dermatologic HIFU platforms, offering four handpieces with nominal focal depths of 0.8, 1.3, 1.8, and 2.3 mm. Typical pulse settings for AK are 0.6–1.2 J for 150 ms (“Superficial” or “Upper-dermis” modes), delivered in a grid pattern 1 mm apart [29]. Real-time video-dermoscopy integrated in the probe facilitates shot to shot placement and immediate endpoint visualization (whitening or pinpoint bleeding) [29].

In an open-label exploratory study of 20 MHz HIFU for AK, Serup et al. followed 201 lesions in eight patients for up to one year and documented a 97% clinical clearance after one to three treatment sessions, with re-epithelialization in 1–2 weeks and no scarring [29]. The protocol employed ultra-short (150 ms) acoustic bursts delivering 0.6–1.2 J per pulse to a 0.1–0.3 mm^3^ focal spot centered 1.3 mm beneath the stratum corneum, thereby confining thermocoagulation to the epidermis/superficial dermis. Immediate pain was described as a brief sting (median visual analog scale [VAS] 4; range 1–8) and was uniformly rated as less intense than prior photodynamic therapy [29]. Transient erythema and superficial crusts constituted the only common adverse effects, resolving spontaneously within two weeks.

A Swiss single-center, prospective cohort enrolled 21 elderly patients (mean age ≈72 years) harboring 108 actinic keratoses (AKs, Olsen grades I–III) and treated them with dermatologic high-frequency high-intensity focused ultrasound (20 MHz HIFU) [73]. Three months after therapy, 72.2% of lesions showed complete clinical clearance and a further 26% exhibited at least one-grade downgrading, yielding an overall objective response of 98%. Pain was brief but bothersome, while late adverse effects (transient hypo-/hyperpigmentation or erythema) occurred in only a minority of sites [73]. All procedures were carried out with the CE-marked System ONE-M™ 20 MHz dermatologic HIFU platform (TOOsonix, Denmark). For each actinic keratosis, the operators selected the 1.3 mm focal depth transducer and delivered single 150 ms ultrasound bursts at 0.9–1.2 J [73].

Histological and high-frequency ultrasound mapping demonstrate concentric thermal necrosis limited to the intended depth, sparing adnexal structures and enabling dense, overlapping passes across chronically sun-damaged skin [74,75]. Lesion-wide “stamping” can be completed in minutes, offering a feasible alternative to topical 5-FU, cryotherapy or daylight-PDT for patients who cannot tolerate prolonged erythema, ulceration, or photosensitivity [66]. From a practical standpoint, employing 20 MHz HIFU to uniformly treat the entire facial field in patients with multiple AKs is essentially unfeasible, whereas true field-based therapies—such as full-face 5-ALA photodynamic therapy (both conventional and daylight-PDT), ablative or fractional laser resurfacing, and medium-depth TCA chemical peels—are routinely deployed to treat extensive photodamaged skin, all of which are supported by clinical studies and endorsed in contemporary European guidelines [71,76,77,78].

Pain during each pulse is typically described as a brief stinging sensation (mean VAS 3–5) and is consistently lower than with conventional photodynamic therapy (PDT) [29,73]. The most commonly reported adverse effects include transient erythema, pinpoint crusting, and focal hypo- or hyperpigmentation. Importantly, no cases of permanent scarring or infection have been documented to date [29,73].

The favorable safety profile of high-frequency HIFU can be attributed to its precise, superficial energy delivery, with limited penetration depth and narrow focal diameter. This anatomical specificity minimizes the risk of damage to deeper structures. A recent study utilizing in vivo modeling of granulomatous tissue further supports this, demonstrating that energy deposition remains confined to the upper dermis with negligible impact on surrounding tissues [61]. HIFU operating at ~20 MHz produces sharply delimited thermocoagulation zones (≈65–80 °C) measuring only 0.1–0.3 mm^3^ at user-selectable focal depths of 0.8–2.3 mm, thereby ablating dysplastic keratinocytes while preserving the overlying stratum corneum [29].

High-intensity focused ultrasound (HIFU) offers several potential advantages in dermatologic applications. The technique enables selective intradermal ablation while preserving the epidermal barrier, contributing to favorable healing outcomes [29,73]. Patients typically experience rapid re-epithelialization within 1–2 weeks, with minimal downtime [29,73]. Unlike photodynamic therapy (PDT), HIFU does not require photosensitization and can be safely applied to hair-bearing areas, such as the scalp [76]. Moreover, it is associated with lower procedural pain compared to PDT and avoids complications such as cold-induced dyschromia [29,73,76].

However, several limitations and unknowns remain. Device availability is currently restricted, and initial capital investment may be significant. The technique also requires an operator learning curve to achieve optimal dosing and treatment precision. Importantly, there is a lack of long-term (>5 years) recurrence data, which limits current understanding of its durability. Finally, robust randomized head-to-head trials comparing HIFU to established modalities are still lacking, underscoring the need for further clinical validation.

Current European guidelines emphasize combined lesion- and field-directed therapies for AK [71]. Hence, 20 MHz HIFU could replace surgery, cryosurgery or lasers for isolated grade I–III AKs unsuitable for topical therapy, rescue lesions unresponsive to PDT or imiquimod, and serve as a rapid, office-based small field treatment when adherence to multi-week topical regimens is unlikely.

Future research should prioritize adequately powered, randomized, dose-finding controlled trials—stratified by Olsen grade, anatomical site, and anticoagulant status—to benchmark 20 MHz dermatologic HIFU against established modalities such as liquid nitrogen cryotherapy, ablative or fractional CO_2_/Er:YAG laser resurfacing, and both conventional and daylight 5-ALA photodynamic therapy for actinic keratosis. Parallel investigations should also determine whether HIFU can serve as an effective salvage option for lesions that remain refractory to photodynamic therapy or other standard field-directed treatments. Incorporating high-frequency ultrasonography (HFUS) into post-treatment assessment could furnish real-time, submillimeter information on residual epidermal and dermal involvement [79], enabling operators to personalize subsequent HIFU passes by modulating pulse width, focal depth, and acoustic energy to accommodate variations in scalp thickness, dorsal-hand laxity or periocular skin, thereby maximizing efficacy while limiting collateral thermal exposure [80]. Field-directed protocols should further quantify subclinical clearance with reflectance confocal microscopy or HFUS and correlate these imaging outcomes with genomic UV-signature burden to refine biomarkers of response [79,80,81,82]. Finally, comprehensive cost-effectiveness analyses—encompassing single-visit procedural efficiency, the absence of dressings, and reduced caregiver time—are needed to delineate real-world adoption thresholds for 20 MHz HIFU. 

The effects of high-frequency HIFU in the treatment of actinic keratosis are illustrated in Figure 3**,** which shows representative clinical and dermoscopic outcomes obtained by the authors.

## 5. HIFU for Non-Melanoma Skin Cancers

### 5.1. Basal Cell Carcinoma (BCC)

Basal cell carcinoma is the most prevalent skin cancer, and while it is generally locally destructive with a low metastatic risk, standard treatments like surgical excision, curettage with electrodesiccation, cryotherapy, or Mohs micrographic surgery can be limited by cosmetic or functional morbidity, especially for lesions on the face [83,84].

There is growing interest in less invasive modalities that can eradicate BCC while preserving healthy tissue and cosmetic appearance [85]. Minimally invasive approaches—including curettage with or without electrodesiccation and cryotherapy, photodynamic therapy, and topical preparations such as 5-fluorouracil, imiquimod, and ingenolmebutate—offer reasonable clearance for low-risk basal cell carcinomas [85]. Their clinical uptake is constrained by the following factors: operator dependency and the requirement for meticulous patient adherence; procedure-related pain combined with only modest long-term efficacy; and confinement to superficial lesions that necessitate prolonged application regimens and carry variable recurrence rates [85].

High-intensity focused ultrasound, with its ability to concentrate ablative energy at precise skin depths, offers a potential alternative to treat BCC lesions without incisions or significant scarring [8].

Dermatologic high-frequency HIFU (∼20 MHz) produces sharply confined thermal coagulation zones (~60–80 °C) of about 0.1–0.3 mm^3^ at controlled focal depths [86]. This allows selective intradermal ablation of tumor tissue while sparing the epidermis and deeper structures. The HIFU device (e.g., System ONE-M, TOOsonix) provides interchangeable transducers with focal depths of 0.8, 1.3, 1.8, and 2.3 mm to match lesion thickness [86]. Superficial BCCs (which typically extend ~1–2 mm deep) are often treated with the 1.3 mm focal depth probe, whereas nodular BCCs up to ~2 mm thick may require the 2.3 mm focus [86]. Each pulse (commonly 150 ms) delivers ~0.7–1.2 J of acoustic energy, inducing coagulative necrosis at the focal point without significant lateral heat spread [86]. In practice, operators can “stamp” the HIFU pulses in a grid across the lesion, and in some cases a “sandwich” technique (sequential HIFU passes at two different depths) is used to ensure full-depth ablation of thicker tumors [29]. A built-in video dermatoscope in the handpiece provides real-time guidance for precise shot placement and immediate visualization of treatment endpoints (such as blanching or pinpoint bleeding) [86]. This allows in situ adjustment of energy delivery and confirmation that the entire clinical tumor area has been covered.

Emerging studies have demonstrated high clearance rates of BCC with 20 MHz HIFU, though data are still limited to small cohorts or case reports only:

Serup et al. presented the first exploratory application of high-frequency (20 MHz) high-intensity focused ultrasound (HIFU) for cutaneous tumors [29]. A single patient with seven basal cell carcinomas that had recurred after photodynamic therapy received one HIFU treatment per lesion, consisting of 150 ms pulses at 1.2 J delivered to a focal depth of 1.7 mm. At 6-month follow-up, all lesions showed complete clinical resolution, and biopsy confirmed the absence of residual carcinoma. Procedure-related pain was moderate (visual analog scale 1–7; mean = 5). These observations provide the first in vivo evidence that 20 MHz HIFU can eradicate basal cell carcinoma with minimal morbidity.

Calik et al. conducted a prospective pilot study evaluating a 20 MHz dermatologic HIFU platform in eight patients (15 histologically confirmed BCC lesions) [8]. Lesion depth was quantified with high-frequency ultrasound before treatment, guiding selection of a focal probe between 0.8 mm and 1.8 mm. Each focal spot received 1.3 J delivered over a 150 ms pulse [8]. Immediate whitening and perifocal edema signaled adequate thermal coagulation, while overlying crusts separated within approximately two weeks, with minimal procedure-related discomfort. At six-month follow-up, every treated site had completely re-epithelialized and blended with adjacent skin, and no clinical recurrences were detected. Patient-reported outcomes were favorable: 75% rated their satisfaction as “very high,” citing negligible scarring and the non-invasive nature of the intervention [8]. Although limited by a small sample size, a short observation period, and the absence of a control arm, these data suggest that dermatologic HIFU can achieve short-term clearance rates for BCC that is on par with surgical excision while potentially conferring superior cosmetic results. Larger, controlled trials with extended follow-up are warranted to confirm durability of response and to refine optimal energy-delivery parameters.

Serup et al. described a 66-year-old woman who had received extensive Grenz-ray irradiation of the scalp in the 1980s and, despite >30 subsequent Mohs, cryotherapy, and curettage procedures, she continued to develop multiple basal cell carcinomas (BCCs) within a “field-cancerized” scalp [86]. Each shot delivered 0.9 J over 150 ms, and 20–200 shots were applied per lesion in accordance with surface area [86]. Using a dermatologic 20 MHz HIFU platform (System ONE-M, TOOsonix) fitted with the 1.3 mm focal depth transducer, the authors ablated eight superficial BCCs and two actinic keratoses during a single outpatient session [86]. Because the irradiated scalp was hyperalgesic, lesions were infiltrated with lidocaine plus adrenaline before treatment [86]. All wounds re-epithelialized spontaneously within approximately three weeks. No dressings or adjunctive therapy were required, and no procedure-related complications occurred [86]. Clinical follow-up at 6–12 months confirmed complete clearance of all eight BCCs. One lesion recurred at month 15, yielding an 87.5% lesion-based one-year cure rate [86]. The authors emphasized that high-frequency HIFU enabled efficient, minimally invasive “field reset”, simultaneously eradicating invasive tumors and surrounding actinic damage with negligible morbidity, and may therefore represent a pragmatic option for patients who have exhausted conventional modalities [86].

Calik et al. described a 68-year-old woman with Gorlin–Goltz syndrome who had developed giant, multifocal, infiltrative basal cell carcinomas (BCCs) involving the abdomen, face, neck, and back, rendering the disease surgically inoperable at presentation [87]. The patient first received conformal external beam radiotherapy (20 Gy in five fractions) followed by, from April 2022 to December 2023, daily vismodegib doses of 150 mg; this sequential approach produced only partial tumor regression and the largest abdominal mass eventually progressed [87]. An R0 excision of the abdominal lesion (bone, abdominal wall, and omentum) was undertaken in December 2023, with subsequent split-thickness grafting for wound closure [87]. Dermatoscopy and 20 MHz ultrasound Dermascan^®^ C (Cortex Technology ApS, Aalborg, Denmark) identified multiple persistent superficial BCCs on the back (depth 0.4–0.8 mm) despite systemic therapy [87]. The patient was then recruited to the prospective Polish trial NCT05698706 evaluating dermatologic high-intensity focused ultrasound (HIFU) for BCC (TOOsonixSystem ONE-M, 20 MHz, 1.3 mm focus) [87,88]. Each pulse delivered 0.7–1.3 J over 150 ms; exposures were laid contiguously with 1–2 mm center-to-center spacing at 1–2 s intervals, ensuring a ~1 mm safety margin around every lesion [87]. All treated sites blanched immediately, re-epithelialized spontaneously within three weeks, and required neither dressings nor analgesics; no acute or delayed adverse effects were recorded [87]. A dermatoscopic review conducted six months later confirmed complete remission of every HIFU-ablated focus, with excellent cosmetic results and no evidence of recurrence or new BCCs [87]. The authors emphasized that high-frequency (20 MHz) HIFU can serve as a precise, minimally invasive treatment for residual or vismodegib-resistant lesions, obviating further systemic therapy and preserving quality of life [87]. High-frequency (20 MHz) HIFU provided rapid, scar-sparing eradication of superficial Gorlin-associated BCCs that had persisted after long-term vismodegib, illustrating its value as a low-morbidity salvage option within truly multimodal skin cancer care.

Safety, Tolerability, and Cosmetic Outcomes: Across studies, HIFU has shown a favorable safety profile in BCC treatment. Patients typically experience a brief, prickling pain during each pulse, but this discomfort is transient and usually does not require local anesthesia. In quantitative assessments, pain scores during 20 MHz HIFU were in the mild to moderate range (e.g., median VAS ~3–5) and were uniformly lower than pain experienced during conventional photodynamic therapy [29]. Post-treatment, an inflammatory response with mild erythema and edema is common, followed by the formation of a thin superficial crust over the ablated zone. These crusts generally slough off within 1–2 weeks as new epidermis regenerates underneath [8]. Importantly, the epidermal barrier disruption is minimal (since the stratum corneum overlying each HIFU thermal zone often remains intact initially), resulting in low infection risk and rapid healing. Significant scarring is either rare or absent, and clinical follow-ups and dermoscopic analyses show that treated sites typically return to almost normal skin appearance or a faint hypo-/hyperpigmented macule after a few weeks to months [8]. Unlike surgery or cryosurgery, HIFU causes no full-thickness wounds and thus preserves skin integrity and adnexal structures, preventing significant alopecia or contractures even in sensitive areas like the scalp or face [86]. No serious adverse events (such as infections, persistent ulceration, or hypertrophic scarring) have been reported in the above studies [8,29,86,87]. Minor transient pigmentary changes have been observed in a minority of cases but tend to resolve over time [86]. Overall, HIFU offers a tissue-sparing approach that maximizes cosmetic outcomes and patient comfort in the management of BCC.

Position in Therapy: Although HIFU for BCC is not yet included in clinical guidelines, the early evidence suggests it could become a useful option in specific scenarios. At present, surgical excision with histological margin control remains the gold standard for BCC treatment, especially in high-risk lesions, given its well established high cure rates [89]. Established nonsurgical alternatives—e.g., topical imiquimod or 5-FU for superficial BCC, or radiotherapy and vismodegib (a Hedgehog pathway inhibitor) for advanced cases—are available, but each has limitations in terms of side effects or cure rate [85]. HIFU may fill an important niche for low- to moderate-risk BCCs in patients who cannot undergo surgery or who have lesions in cosmetically important locations (the face, apart from the eyelids), where even a well-executed excision might cause disfigurement [90]. It may also serve as a “rescue” therapy for tumors that persist or recur after standard treatments like PDT, or in patients with multiple lesions requiring repeated interventions like patients with Gorlin–Goltz syndrome [87]. The ability to treat multiple BCCs in one session is a practical advantage of HIFU, potentially reducing the overall treatment time and burden for patients with syndromes or field cancerization (e.g., basal cell nevus syndrome or chronic radiodermatitis) [86,87]. On the other hand, HIFU equipment is capital-intensive and not widely available outside of specialized centers, and operators require training to optimize energy settings and ensure complete coverage of the lesion. The current clinical evidence is limited to small studies with relatively short follow-up periods, so questions remain regarding long-term recurrence rates and comparative efficacy versus gold standard therapies. For example, in the largest series so far (15 lesions), follow-up was only conducted after 6 months, and in the case involving the difficult scalp, one recurrence did emerge after 15 months [8,86]. The clinical outcomes for BCCs and AKs are summarized in Table 1. Recognizing these gaps, further research is underway—including larger clinical trials—to validate HIFU’s oncologic outcomes [88,91]. Early results are encouraging, and if sustained long-term, 20 MHz HIFU could become an attractive addition to the BCC treatment armamentarium, offering an office-based, cosmetically superior alternative for select patients. In summary, HIFU for BCC is a promising innovation that may complement existing therapies by improving patient comfort and cosmetic results without compromising efficacy [8,86].

The clinical and dermoscopic outcomes of high-frequency HIFU in a histopathologically confirmed basal cell carcinoma are presented in Figure 4, showing the lesion before and after treatment.

### 5.2. Squamous Cell Carcinoma (SCC)

Cutaneous squamous cell carcinoma is the second-most common skin cancer and, in contrast to BCC, carries a significant risk of metastasis if not adequately treated [92]. Standard management of invasive cSCC is surgical excision with histological margin assessment, aiming for complete tumor removal (R0 resection) on the first attempt [92]. For high-risk lesions, Mohs micrographic surgery or wide excision with ≥6 mm margins are recommended to ensure clearance, yielding 5-year cure rates often >90% [92]. Nonsurgical treatments—such as cryotherapy, radiation therapy, or topical agents—are generally reserved for in situ SCC or for patients who are poor surgical candidates, and even then, with caution [92]. This emphasis on surgical management reflects the fact that even relatively small invasive SCCs can recur or metastasize if any portion of the lesion is left untreated. In short, there are few exceptions to the principle that cSCC is best managed with excisional surgery, because reliable clearance of all malignant cells is critical to patient outcomes [92].

Current Evidence and Challenges for HIFU in SCC Treatment: Given the above, the use of high-frequency HIFU in cutaneous SCC remains exploratory. There are currently no published clinical trials or cohort studies specifically examining HIFU for the treatment of invasive SCC of the skin. In theory, the ability of HIFU to precisely ablate tissue at controlled depths could be applied to SCC lesions just as with BCC or actinic keratoses. For example, an in situ SCC (Bowen’s disease) confined to the epidermis or a superficial well-differentiated SCC could potentially be ablated by HIFU energy focused at 0.8–1.3 mm depth, eradicating the neoplastic keratinocytes while sparing the surrounding dermis. The potential advantages would be similar to those in BCC: a non-invasive outpatient procedure with minimal scarring and the ability to treat multiple foci in a field. Indeed, in the previously mentioned case of post-radiation field cancerization, two actinic keratoses, or SCC in situ on the scalp were treated with HIFU alongside BCCs, and those lesions also showed complete clearance at 12 months [86]. This anecdotal finding suggests HIFU can destroy dysplastic squamous lesions in principle. Furthermore, HIFU has been successfully used in other organ systems to treat squamous cell carcinomas (for instance, palliative treatment of esophageal SCCs with focused ultrasound), indicating that SCC tissue is susceptible to ultrasound-induced coagulative necrosis [93].

However, treating an invasive cutaneous SCC with HIFU outside of a trial setting is currently not the standard of care and should be approached with great caution. The key concern is ensuring complete tumor eradication. SCC often extends microscopically beyond what is visible or palpable, and it can infiltrate deeper structures (papillary and reticular dermis, or even subcutis) in a way that might not be uniform [94]. If HIFU energy is delivered only to a fixed depth (e.g., 1.3 mm) and the SCC has portions extending deeper into the skin, there is a risk of under-treatment, leaving the viable tumor at the base or periphery of the lesion. Unlike BCC, which very rarely metastasizes, an incompletely treated SCC poses a risk of metastasis to regional lymph nodes or beyond. Therefore, any HIFU application for invasive SCC would likely need to incorporate histological confirmation of clearance, such as pre- and post-HIFU biopsies of the treated site. There is a possible role for HIFU in SCC as an adjunct; for instance, using HIFU to debulk a thick SCC tumor non-invasively, and then performing a more limited excision or targeted biopsy of the base to ensure no residual carcinoma remains. Another potential application is in palliative scenarios—for patients with unresectable SCC or multiple cutaneous tumors—where standard surgery or radiation is not feasible; HIFU could be explored to control local disease or ulceration. These concepts remain speculative until clinical data are available. To date, no clinical trials evaluating the use of high-intensity focused ultrasound (HIFU) for the treatment of SCC have been registered.

In summary, HIFU for non-melanoma skin cancers beyond BCC (i.e., for SCC and its precursors) is an area in need of further research. The technology’s success in ablating AKs and BCCs provides a rationale that superficial SCC or SCC in situ could likewise be treated if confined to the upper dermis. HIFU might in the future play a role in field-directed therapy for extensive actinic damage to prevent progression to SCC, by treating subclinical clusters of atypical keratinocytes in high-risk chronically sun-exposed skin (analogous to its use in the Grenz-ray case to clean up a field of cancerization) [86]. Nevertheless, until robust evidence and safety data are collected, surgical excision remains the cornerstone for invasive SCC management [92]. Any use of HIFU for SCC at present should be restricted to clinical trials or carefully selected compassionate use, with stringent follow-up. Future investigations will need to establish the efficacy of HIFU in achieving complete pathological clearance of SCC and the long-term recurrence rates compared to surgery. They should also clarify which subsets of SCC (by size, depth, differentiation, or anatomic site) might be suitable for HIFU. Until then, HIFU’s role in SCC should be viewed as an intriguing possibility rather than an established option, in contrast to its growing evidence base in BCC. Continued advances in high-frequency ultrasound technology and non-invasive imaging (to delineate tumor margins and confirm treatment response) could eventually make HIFU a valuable tool in the multidisciplinary management of skin cancer, combining curative intent with excellent cosmetic outcome for carefully selected non-melanoma skin cancers [86].

## 6. Discussion

High-intensity focused ultrasound (HIFU) has recently emerged as a novel non-invasive modality for managing keratinocyte skin cancers and precancerous lesions, with early studies demonstrating promising efficacy in both basal cell carcinoma (BCC) and actinic keratosis (AK) [8,29,73]. In patients with AK, 20 MHz HIFU achieved high clearance rates; one exploratory study reported that ~95% of AK lesions healed with good cosmetic outcome^2^, while another prospective cohort noted 72.2% of lesions completely resolved (and 26% partially responded) at 3 months post-treatment [73]. These outcomes are on par with or superior to conventional therapies, and importantly, HIFU-treated sites showed only short-lived procedural pain, an acceptable healing process, and minimal residual pigmentation changes [73].

For BCC, initial clinical results have likewise been favorable: a pilot study treating 15 BCCs with 20 MHz HIFU observed complete ablation of all lesions with no recurrences during a 6-month follow-up, along with excellent cosmetic results that included rapid re-epithelialization within 2 weeks and minimal scarring [8]. Patients in this series reported high satisfaction, chiefly owing to the procedure’s non-invasiveness and favorable aesthetic outcomes [8]. Although data on squamous cell carcinoma (SCC) are still limited, the successful eradication of intra-epidermal neoplasia (AK) and superficial BCC by HIFU suggests its potential applicability to SCC in situ, pending further investigation. Cautious interpretation is warranted given SCC’s greater metastatic risk, and complete tumor destruction must be confirmed before HIFU can be embraced for SCC treatment.

When comparing HIFU to other nonsurgical therapies, several distinctions become apparent. Photodynamic therapy (PDT) is an established field therapy for AK and superficial BCC, but it requires lengthy clinic visits and is notorious for causing intense pain and post-treatment inflammation. In contrast, HIFU’s treatment course is brief and focal: no photosensitizer or prolonged light exposure is needed, and patients uniformly report significantly less and shorter-lived pain with HIFU than with prior PDT [29]. The acute pain from HIFU occurs only during each ultrasound pulse and subsides quickly, without the protracted burning pain that PDT can induce hours to days after illumination [29,95]. Additionally, HIFU does not carry the photosensitivity precautions needed after PDT and imposes less resource demand on clinics (e.g., no need for special lamps or prolonged chair time) [29,96]. These practical advantages indicate that high-intensity focused ultrasound (HIFU) may serve as an adjunct to PDT for incompletely treated actinic keratoses within areas of field cancerization. At the same time, it must be acknowledged that PDT has a long track record and well-characterized outcomes, whereas HIFU’s long-term efficacy remains to be proven beyond initial follow-ups.

Cryotherapy and topical agents are other widely used, non-invasive treatments for premalignant and superficial malignant lesions. Cryosurgery with liquid nitrogen is a simple, rapid technique suitable for a few isolated AKs or small superficial BCCs, but it is operator-dependent and offers limited depth control. HIFU, by comparison, involves more complex technology and setup—dedicated ultrasound devices and required training—making it more resource-intensive than routine cryotherapy [29,97].

For patients with only a handful of lesions, well-placed cryotherapy or a course of topical field therapy (e.g., 5-fluorouracil or imiquimod) may suffice [29,85].

However, many elderly, fair-skinned patients present with field cancerization (dozens of AKs and incipient BCC/SCC in a sun-exposed region) [29]. In such cases, repetitive cryotherapy is impractical and often ineffective at field clearance, and topical regimens require weeks of adherence with significant skin irritation. HIFU offers a compelling alternative for these patients: multiple premalignant or low-risk malignant lesions can be treated in one session, in virtually any anatomical location, with a controlled depth of thermal ablation for each focus [86]. Under visual guidance, the HIFU handpiece can systematically deliver pulses “shoulder to shoulder” over a broad area, thereby addressing both clinically evident lesions and potentially subclinical extensions in the field [86]. This approach parallels the “field therapy” concept of PDT or 5-FU, but with immediate lesion destruction rather than reliance on patient immune response or chemical cytotoxicity. An added benefit of HIFU is the minimal post-treatment wound care; the induced superficial crusts typically heal within 1–2 weeks without the need for extensive dressings [29]. In contrast, extensive cryotherapy can lead to weeks of erosion and risk of infection, and topical treatments often result in exuberant inflammatory reactions that are bothersome to patients [98]. It is important to note that HIFU does not preclude combination with such modalities. For example, one could envision using fractional laser or curettage to remove thick keratotic scales, then immediately applying HIFU to the fresh AK bases to ensure adequate ultrasound penetration (as thick scales can impede the ultrasound) [29]. Overall, HIFU appears particularly useful for patients with numerous lesions or those who have failed conventional field therapies. Elderly patients with comorbidities—who may be poor surgical candidates and struggle with prolonged topical regimens—stand to benefit from HIFU’s single-session, localized treatment, provided the procedure’s momentary discomfort is managed.

Ablative and fractional laser therapies have also been utilized in the management of superficial NMSCs and extensive photodamage [99,100,101]. Fully ablative lasers (e.g., CO_2_ laser ablation) can effectively vaporize superficial BCCs and AKs and have yielded good cosmetic outcomes in small-case series [100,101]. However, laser ablation, like surgery, lacks real-time depth feedback and cannot easily confirm complete tumor removal, as margins are determined visually, and over-treatment can cause significant scarring while under-treatment risks recurrence [100]. Fractional ablative lasers are generally employed to create microcolumns of injury to rejuvenate photodamaged skin or to enhance drug delivery, rather than to fully eradicate tumors [102]. They have shown promise in improving field therapy; for instance, ablative fractional laser pre-treatment can increase the uptake and efficacy of topical 5-FU or imiquimod in treating AK and superficial BCC [102]. By itself, however, fractional laser therapy usually does not completely eliminate skin cancers. HIFU offers a more selective, tissue-sparing ablation than traditional lasers; the ultrasound focus produces a steep thermal gradient at the lesion’s margins, which helps preserve adjacent healthy tissue [86]. Because most of the acoustic energy is deposited at the focal point and not in the intervening epidermis, the overlying skin remains largely intact (aside from transient coagulative whitening), unlike laser ablation which necessarily disrupts the epidermal surface [86]. This mechanism explains the lack of significant scarring or pigmentary loss after HIFU that has been noted in clinical reports [8,86]. In essence, HIFU can achieve internal destruction of lesions with an “inside-out” healing pattern, whereas lasers cause and “outside-in” injury. That said, lasers and HIFU need not be viewed as mutually exclusive treatments. There may be synergistic opportunities, such as using fractional lasers to pre-treat extensive field areas (to reduce overall actinic damage and stimulate repair) while reserving HIFU for discrete thicker lesions or those requiring precise deeper ablation. Indeed, emerging evidence suggests non-ablative fractional lasers can cut subsequent NMSC incidence by nearly half, by treating photoaged skin [103]; this preventative strategy could conceivably be augmented by focal HIFU ablation of persistent premalignant foci.

Three-dimensional (3D)-guided brachytherapy is another nonsurgical approach that warrants comparison to HIFU, especially in the context of older patients or anatomically challenging tumors. Modern surface brachytherapy uses custom 3D-printed applicators or molds to deliver localized radiation (e.g., HDR brachytherapy) precisely to the skin lesion and thus sparing the surrounding tissue [10]. Clinical studies have reported excellent outcomes with brachytherapy; for example, complete remission of facial BCC/SCC with 95% local control after 5 years, along with superb cosmetic and functional results have been achieved using individualized HDR brachytherapy protocols [9]. Notably, when compared to surgery in sensitive sites (like the nose or eyelid), brachytherapy can offer equivalent tumor control with significantly better patient satisfaction and cosmesis [9]. HIFU shares some of brachytherapy’s advantages in that it is also minimally scarring and can preserve function (no excision of tissue). Both modalities are well-suited for patients who cannot undergo surgery due to frailty or medical contraindications. However, there are practical differences. Brachytherapy typically requires multiple treatment sessions over several days or weeks (to deliver fractions of radiation totaling, say, 40–50 Gy), and it demands access to specialized radiation facilities and expertise in treatment planning. In contrast, HIFU achieves its effect in usually one or two sessions and does not involve ionizing radiation. This means HIFU can be repeated, if necessary, without cumulative dose concerns, which is an important consideration if new lesions arise in the same field. On the other hand, brachytherapy has the benefit of being largely operator-independent once the plan is created (the radiation delivery is uniform and automated), whereas HIFU is quite operator-dependent (the practitioner must accurately target each lesion and decide on energy settings). In terms of acute side effects, HIFU causes an immediate coagulation injury that heals quickly, whereas brachytherapy can lead to transient erythema, desquamation, or chronic telangiectasia in the treated skin. Both approaches generally yield excellent cosmetic outcomes. Directly comparable data are lacking, but anecdotally HIFU-treated sites re-epithelialize with virtually normal skin texture and color [29], and brachytherapy sites often show only slight pigment or hair changes with high rates of patient-rated cosmesis, with >90% reporting “good/excellent” results [9]. From a translational perspective, HIFU might be more accessible in outpatient dermatology clinics (essentially an advanced device-based procedure), whereas brachytherapy is usually confined to oncology centers. In the future, the choice between HIFU and radiotherapy may come down to lesion characteristics and patient preferences: HIFU may be ideal for relatively superficial (<3 mm thick) lesions in patients desiring immediate results without multiple visits, whereas brachytherapy can address larger or infiltrative lesions (including those of >5 mm depth) that HIFU’s current depth of penetration might not fully cover [9,86].

It should also be acknowledged that other ablative modalities, such as electrochemotherapy (ECT) and irreversible electroporation, have been investigated in dermatologic oncology [104,105]. These approaches utilize high-voltage electric pulses to transiently permeabilize tumor cell membranes, thereby enhancing the intracellular delivery of cytotoxic agents [104]. ECT has demonstrated clinical efficacy primarily in advanced or inoperable cutaneous malignancies, including melanoma metastases, locally advanced squamous cell carcinoma, large or recurrent basal cell carcinomas, and cutaneous Kaposi’s sarcomas [104,105]. However, it occupies a distinct therapeutic niche. In contrast, high-frequency (≈20 MHz) dermatologic HIFU is specifically designed for the management of superficial and early-stage lesions confined to the epidermis and upper dermis. Consequently, ECT and other electroporation-based approaches, while promising for the control of late-stage disease, remain beyond the intended clinical scope of dermatologic HIFU addressed in this review.

From a clinical standpoint, HIFU offers several appealing advantages for dermatologic oncology. The procedure is entirely non-invasive—no injections (unless needed for local anesthetic), no incisions, and no need for postoperative wound care—which is particularly advantageous for elderly patients to tolerate HIFU with minimal or no anesthesia, as the acoustic energy is delivered in very short bursts [86]. The ability to treat multiple lesions in one session is another practical benefit in the geriatric population [86]. Cosmetically, HIFU appears to produce outcomes at least as good as the best alternative therapies. Both clinical trials and case reports have documented an absence of significant scarring or dyspigmentation after HIFU, even in delicate areas like the face and scalp [29,73]. Such outcomes represent a major improvement over repetitive surgery, which in chronic NMSC patients can lead to significant cumulative scarring and functional impairment. Early indications also suggest that HIFU yields acceptable recurrence rates in the short to medium term. For example, in a case series of a patient with numerous recalcitrant BCCs, only one out of eight treated tumors recurred within 15 months of HIFU therapy, with all others remaining in remission at one-year follow-up [86]. While this is a limited data point, it demonstrates that durable control is possible; the recurrent lesion in that case was successfully re-treated with HIFU, underscoring the benefit of non-ionizing, repeatable therapy [86]. It should be noted, however that long-term outcomes beyond 1–2 years are virtually unexplored, representing a critical knowledge gap. Most HIFU studies to date have modest sample sizes and follow-ups under 1 year, so true long-term recurrence and any potential late effects are unknown. For instance, a “complete clearance” at 3 or 6 months is encouraging, but NMSC can recur even years later if not fully eradicated. Therefore, extended surveillance of HIFU-treated lesions is necessary to ensure that early efficacy translates into lasting cure rates comparable to surgery and other well established non-invasive methods.

When acknowledging these limitations, one must emphasize that HIFU in dermatology is still in its infancy. The current evidence base, while promising, is composed mainly of pilot studies, small-case series, and short-term follow-ups. There is an inherent publication bias towards positive outcomes in such novel therapies, and the absence of randomized controlled trials means we lack head-to-head comparisons (apart from anecdotal observations) with standard treatments. A call for controlled studies is prudent; for example, a randomized trial of HIFU vs. PDT for extensive AK, or HIFU vs. surgery or cryotherapy for low-risk BCC, would help define relative efficacy, cosmetic results, and patient quality of life. Another limitation is the operator learning curve and technique dependence associated with HIFU. Proper use of this technology requires specialized training: the clinician must understand how to choose appropriate energy settings and focal depths based on lesion characteristics (often determined by high-frequency ultrasound imaging of the tumor) [8]. Especially in treating malignancies, precision is paramount; missing a portion of the tumor due to poor targeting could lead to recurrence. Serup et al. noted that applying HIFU for skin cancers is “more expertise-demanding and more dependent on the operator” compared to treating AK, which is more straightforward [29]. This suggests a need for standardized protocols and possibly image-guidance to improve accuracy (analogous to how brachytherapy or Mohs surgery use mapping to ensure complete coverage). Availability is another practical concern. At present, high-frequency dermatologic HIFU devices are not widely available outside of specialized centers. The technology has only recently been introduced in select university hospitals and clinics in Europe [86]. Cost and infrastructure requirements may limit widespread adoption in the near term. In contrast, nearly every dermatology practice can offer cryotherapy, and radiation oncology centers are equipped for brachytherapy; HIFU will need to demonstrate clear advantages to justify its implementation in routine practice. Lastly, while HIFU avoids many side effects of other treatments, it is not entirely without drawbacks: patients do experience pain during the procedure (albeit briefly), and in some cases local anesthesia or analgesics may be needed for comfort [86]. There is also a possibility of rare adverse effects such as ulceration or infection if excessive energy is delivered, or post-inflammatory pigment changes, although published studies report that these are infrequent and mild [73].

In summary, high-frequency HIFU represents a promising, non-invasive modality for the management of non-melanoma skin cancers and precancerous lesions. Early evidence demonstrates effective lesion clearance, suggesting HIFU’s potential role as both a therapeutic and preventive tool in dermatologic oncology. Nevertheless, the current literature is limited by small sample sizes, short follow-up periods, and heterogeneity in treatment parameters. Future studies should aim to establish standardized protocols, integrate imaging guidance for precise targeting, and explore combinatorial approaches with photodynamic therapy, lasers, or immunomodulatory agents. These perspectives are further elaborated in the Section 7 and Section 7.1.

## 7. Conclusions

High-intensity focused ultrasound is redefining the therapeutic landscape for non-melanoma skin cancers by offering a non-invasive yet effective treatment option that challenges the traditional paradigm of surgical excision. Current evidence indicates that high-intensity focused ultrasound can selectively ablate cutaneous lesions—ranging from benign sebaceous gland hyperplasia and premalignant actinic keratoses to early-stage basal cell carcinomas—and achieve clinical cure rates that are comparable to established therapies while sparing adjacent healthy tissue and yielding excellent cosmetic outcomes [8,37,86]. The absence of incisions and scars, coupled with HIFU’s ability to precisely confine thermal injury to the target lesion, addresses a long-standing need in dermatology for treatments that eradicate cancerous cells without disfiguring the patient [8]. Equally important is the patient’s experience: HIFU is generally well-tolerated, with minimal post-treatment downtime, and it can often be performed in an outpatient setting with only local or no anesthesia, which is especially advantageous for elderly or surgery-averse individuals [86].

In summary, HIFU’s significance in skin cancer management lies in its unique combination of efficacy, selectivity, and patient-centered advantages. It achieves local tumor control through a focused ultrasound beam that produces coagulative necrosis at depth, thereby destroying malignancies in situ while preserving the integrity of the skin’s surface [86]. This mechanism translates into high rates of lesion clearance with minimal cosmetic penalty, as evidenced by the lack of notable scarring or pigment change in reported cases [29]. This technology represents a convergence of therapeutic sensitive areas (face, scalp) and the prioritization of patient quality of life. HIFU is not yet a replacement for surgery in all cases—ongoing surveillance and studies are needed to confirm its long-term cure rates—but it is posited to become an integral part of the current dermatologic practice as a complement to, or in certain cases a substitute for, traditional treatments. In the evolving era of non-invasive skin cancer therapy, HIFU stands out as a modality that can bridge the gap between oncologic control and preservation of form and function, fulfilling the core goal of modern dermatologic oncology: to treat skin cancer effectively without compromising the patient’s comfort or appearance [8].

### 7.1. Future Directions and Perspectives for Skin Cancer Management and Prevention

One exciting avenue for HIFU is its application in field cancerization, treating broad areas of chronically sun-damaged skin to eliminate not only visible lesions but also subclinical precancerous changes. The case of recalcitrant post-radiation BCCs demonstrated that HIFU can be used for field eradication, treating lesions in different stages (AK and BCC) in one session [86]. Future research should explore HIFU protocols for prophylactic treatment of high-risk fields (e.g., chronically sun-exposed scalp or face in immunosuppressed patients) to prevent new skin cancer development. HIFU’s minimal-scarring nature means it could be repeated periodically or used in combination with chemopreventive agents. Additionally, studies could evaluate HIFU vs. traditional field therapies (such as PDT or 5-FU) in reducing the emergence of new NMSCs. Early evidence from laser research shows that improving the health of photodamaged skin (for instance with fractional ablative lasers) significantly lowered subsequent NMSC incidence [103]. HIFU might confer additive benefit by actively destroying incipient tumor foci. This preventive use of HIFU remains speculative but is a compelling concept given the technology’s ability to combine curative and preventative treatment in one procedure [86].

Beyond direct ablation, HIFU may play a role in stimulating anti-tumor immunity. Preclinical studies indicate that HIFU-induced tumor cell death can release tumor antigens and danger signals, functioning as an in situ vaccine that triggers systemic immune responses [106,107]. This immunologic aspect opens the door to combination therapies; for example, using HIFU in conjunction with immune checkpoint inhibitors or other immunotherapies. The rationale is that HIFU could debulk the tumor and simultaneously prime the immune system, thereby improving the ability of systemic therapy to eradicate microscopic disease or prevent metastasis. There is precedent for such synergy in other cancers: focused ultrasound has been shown to enhance the effects of immunotherapy and even produce abscopal (whole-body anti-tumor) effects in experimental models [108,109]. In skin cancer, upcoming trials might evaluate HIFU combined with topical immune modulators (like imiquimod) applied post-HIFU to the treatment zone, aiming to capitalize on the inflammatory milieu created by ultrasound ablation. Another area of interest is HIFU’s potential to influence the tumor microenvironment: sub-lethal HIFU exposures could be investigated for their capacity to modulate cytokine profiles or increase immune cell infiltration in treated fields. Harnessing this immunomodulatory consequence could be particularly beneficial for SCC, which has a higher propensity for invasion; a combined HIFU-immunotherapy approach might improve outcomes in aggressive or recurrent cases that are not ideal for surgery or radiotherapy.

The precision of HIFU treatment can be further enhanced by advanced imaging guidance and artificial intelligence-driven planning. High-resolution ultrasound imaging (20–50 MHz) is already being used to assess lesion depth before HIFU treatment, ensuring that the appropriate focal depth and energy are selected to treat each tumor [8]. In the future, real-time ultrasound or optical coherence tomography (OCT) could be integrated into HIFU devices to allow live visualization of the lesion and immediate confirmation of treatment effects (for instance, seeing the whitening signal of coagulation). Such image-guided HIFU would parallel the MRI- or ultrasound-guided systems used in other fields, adapted for superficial dermatologic use [110]. Additionally, AI-based lesion mapping holds great promise: machine learning algorithms could assist in identifying and delineating clinical and subclinical lesions on imaging or digital photographs, then automate a treatment “grid” for HIFU delivery. For example, AI could analyze a dermoscopic image of a field of AKs, determine which spots require HIFU pulses and at what depth, and then guide the device to treat those lesions with high precision. This would reduce operator dependence and improve reproducibility. AI might also help predict optimal energy settings by analyzing tumor characteristics (thickness, density, anatomical location) on pre-treatment ultrasound, standardizing HIFU dosing for consistent outcomes.

As the repertoire of non-invasive skin cancer treatments grows, a logical progression is to develop combinatorial treatment strategies that leverage the strengths of each modality. HIFU could be combined sequentially or concurrently with other therapies to improve overall efficacy. For instance, one could envision a “HIFU + PDT” hybrid approach. HIFU could target thicker components of a lesion, and PDT could then be applied to the surrounding field to clean up any residual microscopic disease- thus overcoming PDT’s depth limitation while capitalizing on its field effects. Another strategy might be “HIFU + laser adjuvant”, which involves using a fractional laser immediately after HIFU to create channels that facilitate drug delivery into the treated tumor bed. Conversely, performing a fractional laser before HIFU could remove surface hyperkeratosis and enhance ultrasound penetration for more uniform ablation [29]. Topical therapies could also be combined with HIFU in innovative ways. Since HIFU can increase cell membrane permeability via acoustic cavitation and heating, HIFU could be potentially used to drive chemotherapeutic agents or immune stimulants deeper into a lesion (a concept of skin to “sonoporation”) [111,112,113]. For example, low-power pulsed HIFU might enhance the uptake of a topical 5-FU in a thick plaque of actinic keratosis, boosting its effectiveness. Finally, combining HIFU with 3D-printed brachytherapy applicators could be explored for advanced cases; HIFU could debulk large tumors to a smaller volume that can then be more easily covered by a customized brachytherapy dose distribution. Each of these combinatorial approaches will require careful clinical studies to assess safety and benefit. The overarching perspective is that future skin cancer management will likely not rely on a single modality but rather a tailored combination; for instance, mapping by AI, targeted ablation by HIFU, followed by immunotherapy or field therapy to secure long-term control. By integrating HIFU with complementary technologies and treatments, dermatologists can create multi-pronged, patient-specific protocols that maximize tumor clearance while minimizing morbidity.

In conclusion, the horizon for non-invasive skin cancer management is expansive. HIFU’s introduction is part of a broader paradigm shift towards less invasive, image-guided, and tissue-conserving strategies in oncology. As experience grows, we anticipate refinements such as higher-frequency transducers for even shallower focal zones (treating epidermal lesions like carcinoma in situ), portable HIFU devices for clinic use, and standardized treatment algorithms. Preventive applications for treating field cancerization are particularly intriguing and align with the trend of proactive dermatologic care. The convergence of HIFU with imaging, AI, lasers, and immunotherapy represents a future in which clinicians can detect skin cancers earlier (with AI-assisted diagnostics) and treat them instantly and non-invasively. Such a future holds the promise of dramatically reducing the burden of skin cancer by treating lesions with precision and preventing new ones, all while preserving the skin’s appearance and the patient’s quality of life. The next decade of research and clinical innovation will determine how fully this promise can be realized, but current insights clearly point to HIFU as being an important player in the next generation of skin cancer management [8,29,86].

## Figures and Tables

**Figure 1 cancers-17-03518-f001:**
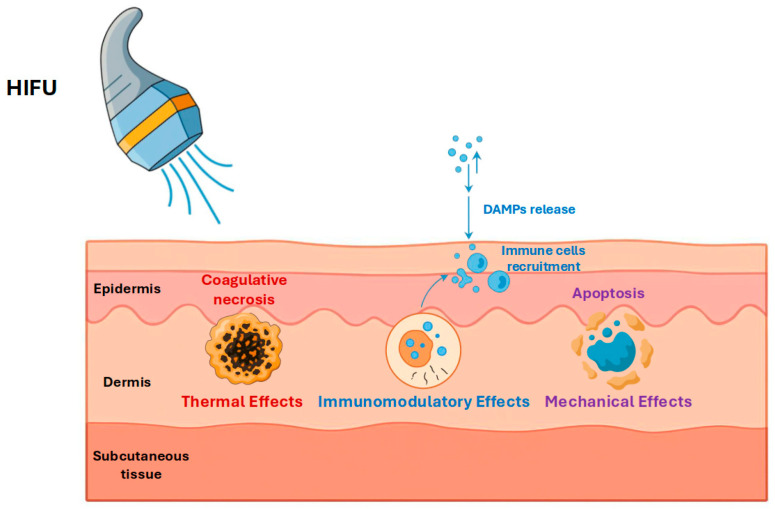
Schematic illustration of HIFU mechanisms of action in dermatology. Thermal coagulative necrosis produced by focused ultrasound (continuous-wave mode), which induces a sharply demarcated zone of tissue ablation at the focal point; acoustic cavitation and mechanical tissue fractionation (*histotripsy*) from pulsed HIFU, resulting in micro-disruption of cells and extracellular matrix; and HIFU-induced immunomodulation, whereby tumor antigens and DAMPs released from ablated cells trigger an anti-tumor immune response (involving cytokine release and T-cell activation) in the host. These synergistic mechanisms allow HIFU to destroy target lesions while sparing surrounding healthy tissue and potentially engaging systemic anti-cancer immunity.

**Figure 2 cancers-17-03518-f002:**
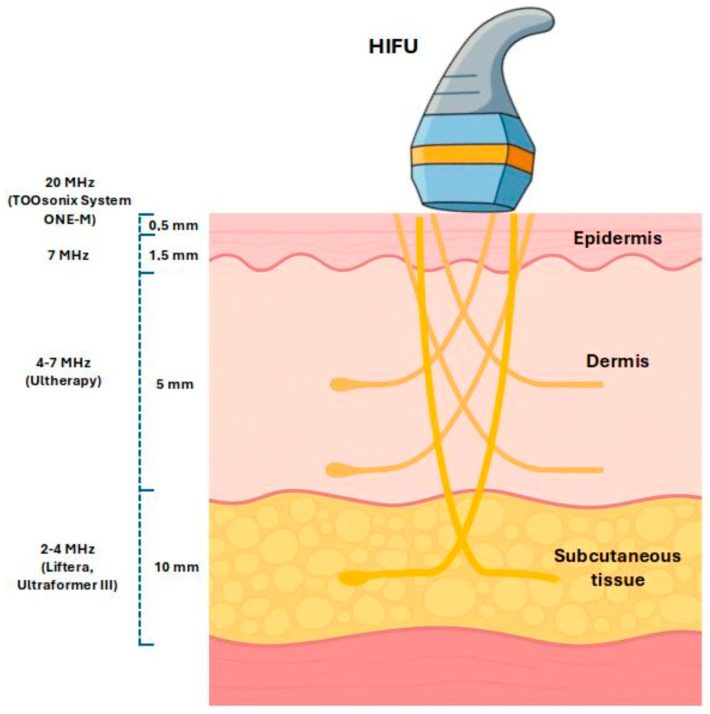
Relationship between ultrasound frequency and depth of penetration for dermatologic HIFU platforms. Higher frequencies (e.g., ~20 MHz) produce very shallow focal zones (approximately 1–3 mm deep) that are appropriate for lesions in the epidermis or superficial dermis, whereas lower frequencies (e.g., 4–7 MHz) can penetrate more deeply (5–13 mm or more) to target subcutaneous structures. Representative commercial devices are operating at the following ranges: Ultraformer III and Liftera (≈2–4 MHz) can reach the subcutaneous fat up to ~10–13 mm for body contouring, Ultherapy (4–7 MHz) treats dermal and SMAS layers at ~3–4.5 mm for facial lifting, and TOOsonix System ONE-M (20 MHz) is designed for superficial lesions in the epidermis and papillary dermis. These device-specific parameters underscore the importance of selecting the appropriate frequency and focal depth based on the lesion’s location and desired treatment outcome.

**Figure 3 cancers-17-03518-f003:**
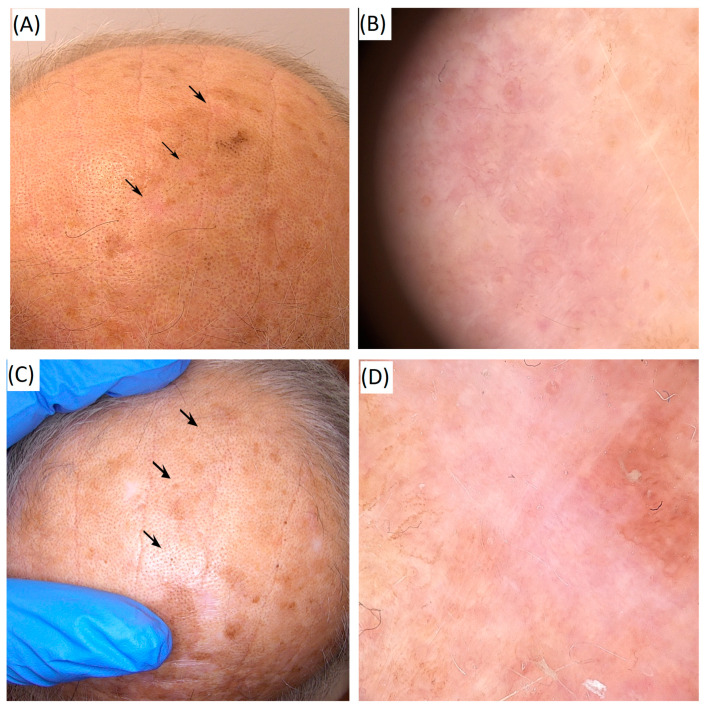
Clinical and dermoscopic images from the author’s collection, published with the patient’s consent. A 68-year-old man is treated with high-intensity focused ultrasound (HIFU) for multiple actinic keratoses and photodamaged scalp skin. (**A**) Clinical image before treatment. (**B**) Dermoscopic image before treatment. (**C**) Clinical image 12 months after a single HIFU session, showing marked improvements in photodamage and lesion clearance. (**D**) Dermoscopic image 12 months after treatment, demonstrating regression of actinic keratoses and improved skin texture, with residual post-treatment hypopigmentation. The arrows indicate the actinic keratosis (AK) lesions before treatment and after treatment.

**Figure 4 cancers-17-03518-f004:**
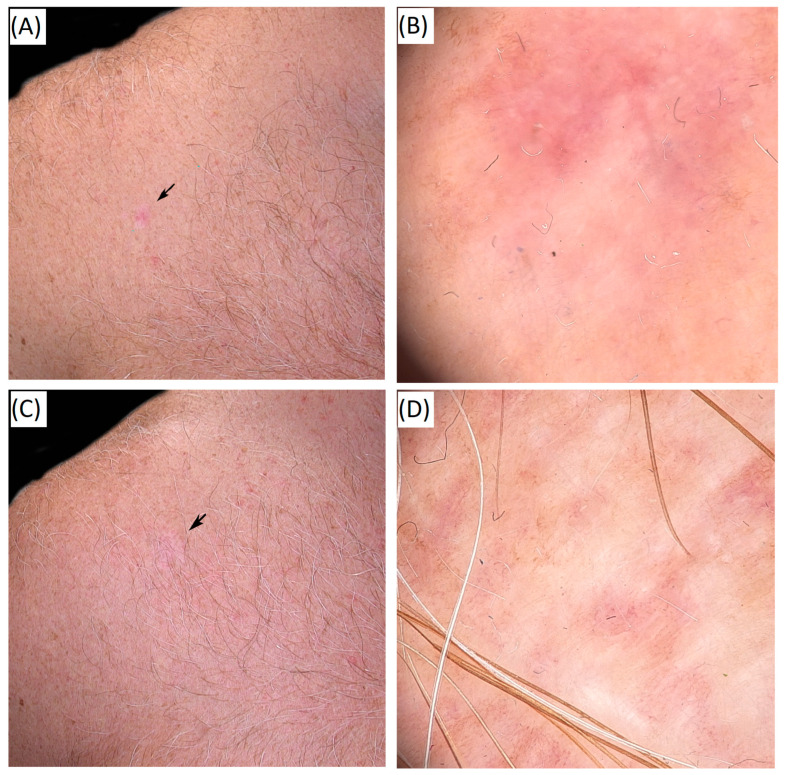
Clinical and dermoscopic images from the author’s collection, published with patient’s consent. A 50-year-old man is treated with high-intensity focused ultrasound (HIFU) for a histopathologically confirmed superficial basal cell carcinoma. (**A**) Clinical image before treatment. (**B**) Dermoscopic image before treatment. (**C**) Clinical image 12 months after treatment. (**D**) Dermoscopic image 12 months after treatment, showing a favorable cosmetic outcome and no evidence of recurrence. The arrows indicate the basal cell carcinoma (BCC) lesions before and after treatment.

**Table 1 cancers-17-03518-t001:** Summary of clinical studies and case reports evaluating HIFU in AKs and BCCs.

Study	Patient Sample	Lesion Type	HIFU Parameters	Clinical Outcomes	Cosmetic Results	Adverse Effects	Follow-Up
Serup et al., 2020 [29]	8 patients with 201 AKs; 1 patient with 7 BCCs	AK, BCC, Kaposisarcoma	20 MHz, 0.6–1.2 J/shot, 150 ms, focal depth 1.3 mm, 1–3 sessions.	97% complete response in AK.	Good with no scar after treatment.	Transient erythema and superficial crusts (resolving within 2 weeks); median VAS pain score: 4.	3–6 months
Seyed Jafari et al., 2022 [73]	21 patients with 108 AKs	AK	20 MHz, 0.9–1.2 J/shot, 150 ms, focal depth 1.3 mm, 1 session.	98.2% total response (72.2% complete, 26.0% partial).	Good	Short pain during the procedure; only a small proportion of lesions showed hypopigmentation, hyperpigmentation, or erythema.	3 months
Calik et al., 2024 [8]	8 patients with 15 BCCs	BCC	20 MHz, 1.3 J/shot, 150 ms, focal depth 0.8–1.8 mm, 1 session.	100% response; no recurrence detected during the follow-up period.	75% of patients reported their satisfaction as “very high”.	Transient erythema, crusting, edema, and short-term urticaria; no infections or significant scarring reported.	6 months
Serup et al., 2024 [86]	1 patient with 8BCCs and 2 AKs	BCC, AK	20 MHz, 0.9 J/shot, 150 ms, focal depth 1.3 mm, 1 session.	87.5% clinical response was observed in BCC; 1 lesion recurring after 15 months.	Good	No procedure-related complications occurred.	6–15 months
Calik et al., 2024 [87]	1 patient with 2 BCCs (associated with Gorlin–Goltz syndrome)	BCC	20 MHz; 0.7–1.3 J/shot; 150 ms pulse duration; focal depth 1.3 mm; single session; combined with vismodegib.	100%	Cosmetic outcome reported as “excellent” by the authors.	No acute or delayed adverse effects were recorded.	6 months

Abbreviations: HIFU, high-intensity focused ultrasound; AK, actinic keratosis; BCC, basal cell carcinoma; VAS, visual analog scale.

## Data Availability

No new data were created or analyzed in this study.

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
