# Peer review of "High-Intensity Focused Ultrasound in Dermatology: A Review with Emphasis on Skin Cancer Management and Prevention"

_cancers, 2025, doi:10.3390/cancers17213518_

Round 1

Reviewer 1 Report

Comments and Suggestions for Authors

The manuscript is a comprehensive review of the basics and clinical applications of HIFU. The authors are exoerts in the field and provide very good overview of the topics, pointing out also of the drawbacks of the tehnology.

What I miss is a bit more about other ablative techniques that are used in treatment of cutaneous malformations or tumors. Mentioned are photodynamic therapy adn some others, but skipped are electroporation based approaches like electrochemotherapy.

Generally, I recommend the manuscript for publication.

Author Response

Thank you very much for taking the time to review this manuscript. Please find the detailed responses below.

Comments 1: What I miss is a bit more about other ablative techniques that are used in treatment of cutaneous malformations or tumors. Mentioned are photodynamic therapy adn some others, but skipped are electroporation based approaches like electrochemotherapy.

Response 1: We thank the reviewer for this valuable suggestion. In response, we have added a new paragraph in the Discussion section (Section 6, page 25) addressing electroporation-based techniques, including electrochemotherapy (ECT) and irreversible electroporation. However, we decided not to expand this topic further, as these modalities are primarily applied to advanced or inoperable cutaneous malignancies, whereas high-frequency dermatologic HIFU is intended for early-stage, superficial lesions. Thus, both techniques target different stages of disease progression and occupy distinct therapeutic niches.

The added paragraph reads as follows:

“It should also be acknowledged that other ablative modalities, such as electroche-motherapy (ECT) and irreversible electroporation, have been investigated in dermato-logic oncology [104, 105]. These approaches employ high-voltage electric pulses to transiently permeabilize tumor cell membranes, thereby enhancing the intracellular delivery of cytotoxic agents [104]. ECT has demonstrated clinical efficacy primarily in advanced or inoperable cutaneous malignancies, including melanoma metastases, locally advanced squamous cell carcinoma, large or recurrent basal cell carcinoma, and cutaneous Kaposi’s sarcoma [104,105]. However, it occupies a distinct therapeutic niche. In contrast, high-frequency (≈20 MHz) dermatologic HIFU is specifically designed for the management of superficial and early-stage lesions confined to the epidermis and upper dermis. Consequently, ECT and other electroporation-based approaches, while promising for the control of late-stage disease, remain beyond the intended clinical scope of dermatologic HIFU addressed in this review.”

Comments 2:  Generally, I recommend the manuscript for publication.

Response 2: Thank you very much for your valuable recommendation.

Additional Clarifications: We have revised the structure of the abstract to improve clarity and coherence. Additionally, we have expanded the manuscript to include the suggested explanations and clarifications, ensuring a more comprehensive discussion  of our findings. Moreover, the attached table has been revised to enhance clarity and overall readability.

Once again, we sincerely appreciate your valuable feedback and constructive suggestions, which have helped us improve the quality of our manuscript.

Best regards,

Bartosz Woźniak

Reviewer 2 Report

Comments and Suggestions for Authors

The authors of High-Intensity Focused Ultrasound in Dermatology: A Review 2 with Emphasis on Skin Cancer Management and Prevention narrative review present integrative analysis of mechanistic insights, clinical outcomes, and comparative positioning within the broader therapeutic landscape of high-intensity focused ultrasound (HIFU). HIFU has recently emerged as a novel non-invasive treatment modality in dermatology, offering precise ablation of cutaneous lesions with minimal damage to surrounding tissue.

This review outlines the mechanistic basis of HIFU- including thermal coagulation, acoustic cavitation, and immunomodulatory effects—and presents the current evidence for its efficacy in treating actinic keratoses and basal cell carcinomas (BCCs), where early 18 studies report clearance rates of 70–97% and excellent cosmetic outcomes. Compared to conventional therapies such as surgery, photodynamic therapy, or cryotherapy, HIFU offers reduced procedural pain, faster healing, and the ability to treat multiple lesions in a single session.

  • In introduction section, at the end, please specify more clearly the aims of this review.

  1. Mechanism of HIFU in dermatology
  2. Current Dermatological Uses of HIFU (Cosmetic and Benign Applications)
  3. HIFU for premalignant lesions (Actinic keratosis and field cancerization)
  4. HIFU for Non-Melanoma Skin Cancers
  • The data from above sections are sustained by 4 illustrative figures.
  • Table 1: please organize the table data according to MDPI format.
  • Kind suggestion for this paper comparison: DOI: 10.3390/app11209511
  1. Discussion

  • Please present more concise information from:
  1. Conclusions
  2. Future directions and perspectives for skin cancer management and prevention

Author Response

Thank you very much for taking the time to review this manuscript. Please find the detailed responses below.

Comments 1: In introduction section, at the end, please specify more clearly the aims of this review.

Response 1: The following sentence has been added at the end of Section 1 (Introduction, page 3):
“The specific aim of this review is to summarize the mechanistic principles, current dermatological applications, and clinical evidence of HIFU, with particular emphasis on its emerging role in the management and prevention of skin cancers.”

Comments 2: HIFU for Non-Melanoma Skin Cancers

  • The data from above sections are sustained by 4 illustrative figures.
  • Table 1: please organize the table data according to MDPI format.
  • Kind suggestion for this paper comparison: DOI: 10.3390/app11209511

Response 2: The revised version of Table 1 has been reformatted according to the MDPI guidelines and is now included in the updated manuscript. We thank the reviewer for the kind suggestion. The recommended article (DOI: 10.3390/app11209511) has been reviewed; however, as it focuses on molecular detection of BRAF mutations in melanoma patients and does not address HIFU or therapeutic ultrasound applications, it was not included in the final version of the manuscript.

Comments 3: 

  1. Discussion
  • Please present more concise information from:
  1. Conclusions
  2. Future directions and perspectives for skin cancer management and prevention

Response 3:  At the end of the Discussion section, we have added a concise summary paragraph that synthesizes the key points from the Conclusions and Future Directions sections to improve clarity and cohesion. The paragraph reads as follows (page 25):

“In summary, high-frequency HIFU represents a promising, non-invasive modality for the management of non-melanoma skin cancers and precancerous lesions. Early evidence demonstrates effective lesion clearance suggesting a potential role as both a therapeutic and preventive tool in dermatologic oncology. Nevertheless, the current literature is limited by small sample sizes, short follow-up periods, and heterogeneity in treatment parameters. Future studies should aim to establish standardized protocols, integrate imaging guidance for precise targeting, and explore combinatorial approaches with photodynamic therapy, lasers, or immunomodulatory agents. These perspectives are further elaborated in the Conclusions and Future Directions sections.”

Once again, we sincerely appreciate your valuable feedback and constructive suggestions, which have helped us improve the quality of our manuscript.

Best regards,
Bartosz Woźniak

Reviewer 3 Report

Comments and Suggestions for Authors

All new medical technologies may have hope and become valuable tools for clinicians, or will be relegated to the category of interesting but not useful devices. The authors propose that their studied family of High-intensity focused ultrasound devices has future utility.

Readers of this manuscript are faced with a "wall of text" with few tables for summarizing information. For each condition in the dermatologic uses section, as is appropriate, summaries of the clinical results in tabular form will be helpful.

Have any randomized controlled trials been performed using this modality compared with others? If not, this entire field is speculative, and this should be clarified. If many have been performed, a summary of these should be presented.

Table 1 is formatted in a way that makes it challenging for readers to extract any information.

Author Response

We thank the reviewer for this valuable feedback.

Comments 1: Readers of this manuscript are faced with a "wall of text" with few tables for summarizing information. For each condition in the dermatologic uses section, as is appropriate, summaries of the clinical results in tabular form will be helpful.

Response 1: We acknowledge the concern regarding the “wall of text” and the need for clear summarization of clinical results. After careful consideration, we decided not to create an additional table for the dermatologic indications, as the available data are highly heterogeneous and primarily derived from case reports, small single-centre prospective studies, or meta-analyses involving a limited number of patients per indication (e.g., viral warts, sebaceous hyperplasia, scars). Given these small and variable datasets, a separate table would not provide meaningful quantitative synthesis and could potentially overemphasize low-level evidence.

Comments 2: Have any randomized controlled trials been performed using this modality compared with others? If not, this entire field is speculative, and this should be clarified. If many have been performed, a summary of these should be presented.

Response 2: We also emphasize that very few prospective or comparative studies—and no randomized controlled trials—have been published to date. This limitation and the speculative nature of early findings have been explicitly discussed in the Discussion section. The relevant excerpt (page 25) reads as follows:

“It should be noted, however, that long-term outcomes beyond 1–2 years are virtually unexplored—a critical knowledge gap. Most HIFU studies to date have modest sample sizes and follow-ups under 1 year, so true long-term recurrence and any potential late effects are unknown. For instance, a ‘complete clearance’ at 3 or 6 months is encouraging, but NMSC can recur even years later if not fully eradicated. Therefore, extended surveillance of HIFU-treated lesions is necessary to ensure that early efficacy translates into lasting cure rates comparable to surgery and other well-established non-invasive methods.
Acknowledging these limitations, one must emphasize that HIFU in dermatology is still in its infancy. The current evidence base, while promising, is composed mainly of pilot studies, small case series, and short-term follow-ups. There is an inherent publication bias towards positive outcomes in such novel therapies, and the absence of randomized controlled trials means we lack head-to-head comparisons (apart from anecdotal observations) with standard treatments. A call for controlled studies is prudent—for example, a randomized trial of HIFU vs. PDT for extensive AK, or HIFU vs. surgery or cryotherapy for low-risk BCC, would help define relative efficacy, cosmetic results, and patient quality of life.”

In summary, we have clarified throughout the revised manuscript that the current evidence remains preliminary, non-comparative, and hypothesis-generating, and that well-designed randomized controlled trials are needed to establish the definitive role of HIFU in dermatologic oncology and benign indications.

Comments 3: Table 1 is formatted in a way that makes it challenging for readers to extract any information.

Response 3: We thank the reviewer for this helpful comment. Table 1 has been reformatted according to the official MDPI style guidelines to improve clarity and readability. The revised version now presents the data in a structured, consistent layout that facilitates comparison across studies and parameters. The updated table can be found on pages 19–20 of the revised manuscript.

Once again, we sincerely appreciate your valuable feedback and constructive suggestions, which have helped us improve the quality of our manuscript.

Best regards,

Bartosz Woźniak

Round 2

Reviewer 3 Report

Comments and Suggestions for Authors

The responses of the authors are reasonable. I accept their expertise in this area.